# Operational Protocols for the Use of Drones in Marine Animal Research

**Vincent Raoult** [1,*] , **Andrew P Colefax** [2] , **Blake M. Allan** [3] , **Daniele Cagnazzi** [4] ,
**Nataly Castelblanco-Martínez** [5,6] , **Daniel Ierodiaconou** [3] , **David W. Johnston** [7] ,
**Sarah Landeo-Yauri** [6] , **Mitchell Lyons** [8] , **Vanessa Pirotta** [9] , **Gail Schofield** [10]
**and Paul A Butcher** [11]

1   School of Environmental and Life Sciences, University of Newcastle, Ourimbah, NSW 2258, Australia
2   Sci-eye, P.O. Box 4202, Goonellabah, NSW 2480, Australia; acolefax@scieye.com.au
3   School of Life and Environmental Sciences, Centre for Integrative Ecology, Deakin University,
    Warrnambool, VIC 3280, Australia; b.allan@deakin.edu.au (B.M.A.);
    daniel.ierodiaconou@deakin.edu.au (D.I.)
4   Marine Ecology Research Centre, School of Environment, Science and Engineering, Southern Cross
    University, P.O. Box 157, Lismore, NSW 2480, Australia; daniele.cagnazzi@scu.edu.au
5   Consejo Nacional de Ciencia y Tecnología/Universidad de Quintana Roo, Chetumal,
    Quintana Roo 77019, Mexico; dncastelblancoma@conacyt.mx
6   Fundación Internacional para la Naturaleza y la Sustentabilidad, Chetumal, Quintana Roo 77019, Mexico;
    sslandeo@gmail.com
7   Division of Marine Science and Conservation, Nicholas School of the Environment Duke University Marine
    Laboratory, 135 Duke Marine Lab Rd., Beaufort, NC 28516, USA; david.johnston@duke.edu
8   Centre for Ecosystem Science, University of New South Wales, Sydney, NSW 2052, Australia;
    mitchell.lyons@unsw.edu.au
9   Marine Predator Research Group, Department of Biological Sciences, Macquarie University,
    Sydney, NSW 2109, Australia; vanessa.pirotta@hdr.mq.edu.au
10  School of Biological and Chemical Sciences, Queen Mary University of London, London E14NS, UK;
    gail.schofield@qmul.ac.uk
11  New South Wales Department of Primary Industries, National Marine Science Centre, Southern Cross
    University, Coffs Harbour, NSW 2450, Australia; paul.butcher@dpi.nsw.gov.au
*   Correspondence: Vincent.raoult@newcastle.edu.au

**Abstract:** The use of drones to study marine animals shows promise for the examination of numerous aspects of their ecology, behaviour, health and movement patterns. However, the responses of some marine phyla to the presence of drones varies broadly, as do the general operational protocols used to study them. Inconsistent methodological approaches could lead to difficulties comparing studies and can call into question the repeatability of research. This review draws on current literature and researchers with a wealth of practical experience to outline the idiosyncrasies of studying various marine taxa with drones. We also outline current best practice for drone operation in marine environments based on the literature and our practical experience in the field. The protocols outlined herein will be of use to researchers interested in incorporating drones as a tool into their research on marine animals and will help form consistent approaches for drone-based studies in the future.

**Keywords:** review; UAV; drones; marine; methods; approach; research; animals; protocols; behaviour

## 1. Overview

The use of drones to study marine animals is increasing as they allow research on the movement, ecology, behaviour, health and habitat use of various marine taxa [1,2]. Since the number of drone-based

studies is rapidly increasing, there are concerns that drones are being used improperly in ways that may invalidate research results, either through inadequate equipment or through incorrect flight protocols that may impact the behaviour of the observed animals [3]. Here we present a synopsis that presents flight protocols on all taxa where drones have been used with success to prevent known errors in flight protocols from becoming commonplace. In general, these protocols require identifying the traits unique to different species, understanding the information that can be made available through drone research, selecting an appropriate drone, understanding any impacts drones can have on the study species, and using appropriate flight patterns to optimize data (Figure 1). In some cases, taxa-specific environmental conditions or data processing may be required. These protocols, as they have all been developed by researchers with considerable experience using drones for specific taxa, should be considered best practice for researchers either using drones currently or who are considering the use of drones for marine research. We start by presenting broad principles on the use of drones for marine research that will provide readers with current information on terminology, equipment, processing, and environmental aspects of drone-based marine research. We then provide taxa-specific protocols for the use of drones for jellyfish, sharks, reptilians, marine birds, pinnipeds, sirenians, odontocetes, and mysticetes that are typically found close to the surface or on land where drones are effective.

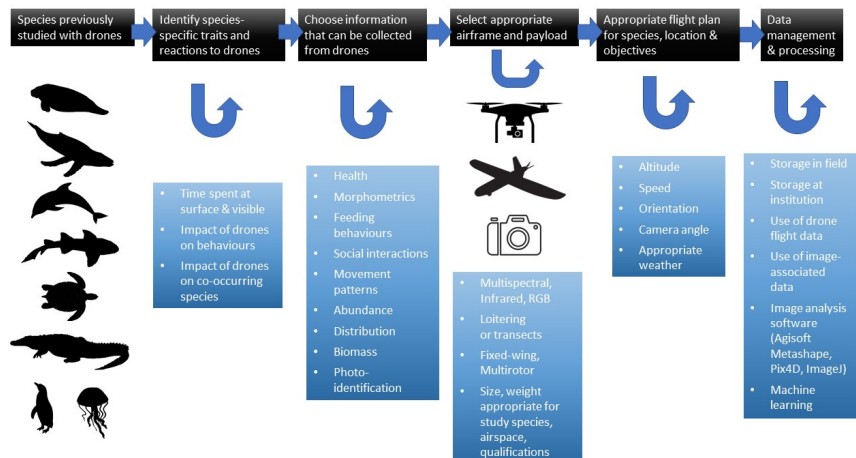

**Figure 1.** Overview of operational protocols used for drone research on marine animals.

## 1.1. Different Drone Platforms Used in Marine Research

Drones come in various shapes and sizes, but the type of platform used is critical to successfully study a chosen taxon, especially with regards to the motorisation of aircraft. Many drone platforms are available to marine researchers, and each has different capabilities and strengths. Single-rotors (helicopters) are older platforms with high lift capacity, endurance, and stability in adverse weather conditions [4]. Single-rotor platforms are now a niche product, as they have no redundancy, complex machinery (collective pitch, tail rotor, petrol engines), higher risk factors from greater disk loading (more energy in rotors), and high rates of vibration that obscure images. Single-rotors are typically only used when heavy lifting is necessary. The most common modern platform is the multirotor, which can have anywhere from 4 to 8 or more motors and rotors. Multirotors have the advantage of having few moving parts, which reduces costs, provides redundancy in some airframes in case of failure [5], are safer to operate than single-rotors, and produce comparatively few vibrations relative to single-rotors. Relying on rotors for lift is notoriously inefficient; single and multirotors have relatively low flight endurance. One key advantage is that multirotors can take off and land in very small areas. In contrast, fixed-wing drones that are essentially flying wings with one or two motors have higher endurance (>90 min) and can cover larger areas as a result, but require larger areas than multirotors to take off and land. All modern drones can have largely autonomous flying

that requires very little input from the pilot to conduct missions. Since each drone platform has its strengths, some are more adapted to research on one animal taxa more than others.

### 1.2. Payloads Available to Drones for Marine Research

Payloads are the packages that the drone platform carries and are used to study organisms. The most common type of payload used in drone research is a digital camera to obtain red-green-blue (RGB) images and video. These can vary from smaller, lower quality lenses found in lower-cost drones to expensive and high-quality lenses or digital single lens reflex systems on larger drones. Higher quality and larger lenses are also typically bulkier and heavier, meaning these payloads are generally restricted to larger drone platforms. In the last five years, the incorporation of infrared and near-infrared payloads has become more commonplace [6,7], although these are generally more expensive than conventional imaging. These allow temperature to be measured for animals visible from the air (not shielded by tree canopy or soil/rocks) [8]. Drones can also carry laser altimeters that are more accurate than normal onboard barometers and can increase the accuracy of photogrammetry and structure-from-motion (SfM) processing [9].

### 1.3. Photogrammetry and Structure-from-Motion from Drones

One of the strengths of drones is that, aside from obtaining visual information like behaviour, photogrammetry (measuring objects from pictures) and SfM (creating 3D models from multiple images) can also be performed [10]. Photogrammetry and structure from motion allow researchers to create maps (orthomosaics) as well as digital surface models from point cloud data [11], allowing the capture of morphometric measurements of animals in the wild without requiring capture, and to make assessments on the health, weight, and demographics (see jellyfish and pinniped sections). The incorporation of numerous sensors including Global Positioning System (GPS) and barometers in most drone platforms also means that photogrammetric measurements obtained from drones can produce measurements with error levels <1 cm, especially if precise ground control points (GCPs) or GLONASS/RTK (Gobal Navigation System/Real-Time Kinematic) systems are incorporated [12]. There are numerous software suites that facilitate the processing of these data (e.g., Pix4D, Agisoft Metashape) available to researchers.

### 1.4. Current Limitations of Drone Technology

Ground sampling distance (GSD) is a key factor that limits the accuracy of photogrammetry, the ease of detecting organisms, and by association can force pilots to fly at altitudes that may negatively impact wildlife. The current generation of consumer-grade drones have ~20 MP cameras, however the next generation of drones are expected to move towards the 8 K format or ~48 MP (e.g., Autel Evo 2 released January 2020), which would more than double current ground sampling distance. By association, this advance may enhance detection of organisms through the addition of more detail, increase the speed of surveys (since flights can occur at double the altitude and cover 4 times the area), increase the precision of photogrammetry, and increase the altitude to reduce negative impacts on wildlife. Drones with zoom lenses achieve some of these objectives (e.g., the DJI Mavic 2 Zoom) but cannot provide the same level of ground-sampling distance and come at a cost of cameras with lower resolution (Figure 2).

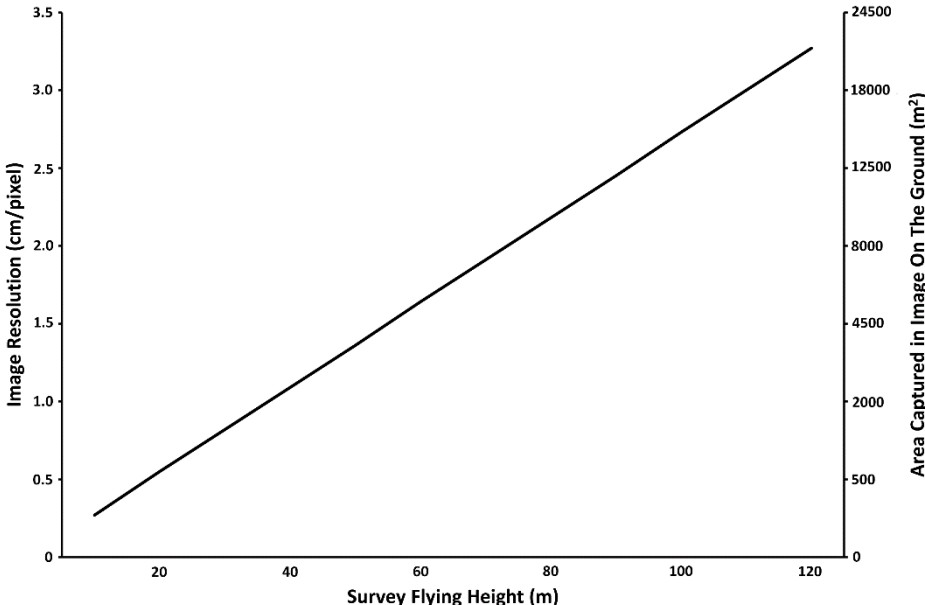

**Figure 2.** Relationship between camera pixel resolution (20MP DJI Phantom 4 Pro), height above ground, image ground sampling resolution (GSD), and the area captured in a single image. Payloads with higher camera resolution have a higher line intercept (higher GSD and area covered) while lower camera resolutions will have a lower line intercept (lower GSD and area covered).

Conversely, as camera resolutions increase and reduce ground sampling distance, data management becomes an ever-increasing problem. At 4K resolution, video produces ~32 GB per hour. In the field, managing the data produced over days or weeks becomes logistically challenging when cloud storage is not available, with multiple external hard drives being required. Researchers currently often balance the need to collect high-resolution data with data management in the field, often resulting in data being recorded at less than optimal resolutions to ensure sufficient storage space is available. These factors will be compounded with increasing camera resolutions and highlight the need for appropriate redundant mobile storage devices in the field. There are also associated issues with long-term data storage from projects that may require specific infrastructure.

Battery technology and the associated limited flight times are another limitation to drone technology. Currently, consumer drones and most multirotors are limited to flight times of <30 min (accounting for a 30% reserve to preserve batteries), while fixed-wing drones are often limited to <90 min. Improvements in battery energy density and drone efficiency have seen almost 50% increase in flight times since the last generation of consumer drones (e.g., DJI Phantom 3 pro = 23 min max, Phantom 4 pro = 30 min max), whereas the next generation of drones is expected to have a further 50% improvement (Autel Evo 2 = 40 min). For monitoring behaviours that may extend beyond battery capacity, drones require battery replacement that interrupt monitoring and make finding the animals of interest again difficult. Over these sorts of flight times, pilot fatigue is likely to become an issue if flights are conducted manually.

*1.5. Considerations for Drone Operations in Marine Environments*

The working environment where drones are used can highlight current technological limitations. Lithium-polymer batteries used in most airframes have operational limits 15–40 °C. This makes their use in cold or hot environments difficult, requiring heated pouches or coolers, respectively, to preserve battery function. In both cases, the use of drones in these environments likely reduces battery efficiency and, therefore, flight endurance. Operating drones in marine environments also exposes aircraft marine aerosols that corrode metallic components both outside and inside the airframe. While some waterproof drones exist that could reduce these effects, these alternatives usually have lower endurance

than normal airframes, and are used less often. When researchers fly repeatedly in areas with high marine aerosol loading (typically beaches), we suggest that the closer monitoring of drone components may be required. Routine monitoring will identify corrosion problems early, and repeated cleaning of airframes after flights using rubbing alcohol (which dries quickly) may be required to prolong the lifespan of components.

In many cases, deploying and retrieving drones occurs from small boats once animals are observed or in remote areas. These are not ideal drone deployment platforms because of electrical and magnetic interference, unpredictable movement of vessels, and the restricted space available. Operational considerations for the operation of drones on boats should start with careful choice of an appropriate drone for boat-based operations. At start up, most drones will calibrate their micro-electro-mechanical system (MEMS) sensors (e.g., accelerometers, gyro), which requires the system to be placed on stable flat surface. These conditions are rarely available during boat-based missions, and researchers can opt to use drones that load previous calibration data created on land. Consideration for the safe launch and recovery of the aircraft is paramount to avoid any magnetic interference on the aircraft (steel hulled vessels for instance) and the combined effect of the vessel and drone movement. Often hand catching provides the safest method to allow for the pitch and roll of the vessel and to keep it at a safe distance from other personnel. This requires precise and clear lines of communication with the skipper and co-ordination. A significant overhead should be maintained with battery voltage to allow for safe and low stress operations, allowing for changes in sea state and wind during the flight. Care should also be taken when programming failsafe modes for drones in these situations. For example, many drones come preprogramed to return to home and land when malfunctioning, or when battery reserves are low. Unfortunately, many systems do not update that home location after launch, which means that the home location may be over water and not an ideal place to land especially if the vessel is underway. Even with a dynamic home position update, the complexities involved in auto-landing on a small vessel are hazardous. Geo-fencing (operational limits from a home point) can also be set by default and effectively prevent drones from returning to a boat that has drifted, and these settings should be disabled. Similarly, many drones do not have object avoidance systems, and return to home fail safes may result in collisions with dynamic or unanticipated components of the local environment (e.g., tall icebergs in high latitude systems or high trees near shore) [13]. Thus, launching drones from moving vessels requires significantly more pilot training and experience than deployments from land, and we recommend pilots only attempt this after being comfortable flying in more benign environments.

Detection of marine animals may be hindered by sun glare/glitter, wind (strong winds >20 km/h reduce stability and ability to maintain path of the drone), turbidity/water clarity (but to a much lesser extent compared to boat-based and underwater surveys), sea conditions (particularly breaking waves), tidal state, awkward body angles [14–16]. Polarizing filters that can reduce reflections, neutral density filters that limit overexposure, and adjusting the angle of the camera between nadir (i.e., 90° from horizon) and 45° forward sometimes helps reduce sun glare. In many cases, operation during mid-day makes it harder to identify animals due to higher chances of sun reflections and no visible shadow for animals, which make them harder to detect [17,18]. Changing the compass orientation of the drone so the camera points away from the sun can also reduce glare. Kelaher, et al. [19] operated the drone manually to maintain the edge of the video footage on the back of the surf break to maximise coverage. Wind speed seems to be more of a limitation for multirotors compared to fixed-wing [20,21]; however, operating drones in winds of 20–25 km/h still provides information that could not be obtained from a boat under the same conditions [16]. Effective use of drones in marine environments thus requires a good understanding of the capabilities of the aircraft as well as how the payload will interact with the current and projected flight conditions.

Most of the species within the taxa evaluated herein are considered Vulnerable or Endangered by the IUCN Red List. Therefore, the development and application of protocols and tools to monitor their numbers are necessary to guide research and management decisions. Based on this general

overview, researchers have provided operational guidelines for key taxa from hundreds of hours of drone operations designed to capture the idiosyncrasies inherent to studying individual taxa to provide some guidance for field ecologists and managers considering adding drones to their toolkit.

## 2. Protocols for the Use of Drones with Specific Taxa

### 2.1. Protocols for the Study of Jellyfish

The study of jellyfish populations poses unique challenges relative to other marine animals. These relate in part to boom and bust patterns that may occur in very short (week to month) timeframes relative to other phyla [22], and their soft bodies that are easily damaged by capture [23]. Due to the increased recognition of the importance of these animals to the food chains of marine ecosystems [24], and their growing value for commercial fisheries, developing effective tools that facilitate the monitoring of these animals will be of benefit to numerous stakeholders.

Drones have already proven effective at assessing the abundance, distribution, and demographics of jellyfish aggregations [25,26]. The ability of drones to rapidly identify small yet dense aggregations across large areas combined with photogrammetry allow managers to frequently and easily assess jellyfish biomass in coastal areas at rates required to effectively monitor jellyfish boom and bust cycles. Effectively using drones for these applications does, however, require appropriate protocols.

2.1.1. Unique Aspects of Jellyfish Protocols

Assessing the spatial distribution or size demographics of jellyfish requires the use of photogrammetry. Unlike scleratinian corals that have also been studied with drones [11], jellyfish Zare not immobile, but neither do they move rapidly. The relative immobility of the subject is a critical component of photogrammetry, without which successive images cannot be aligned. To allow photogrammetric measurements of jellyfish, it is therefore necessary to either capture the entire aggregation in a single image, or to capture multiple successive images with minimal delay between them so that there is little perceivable movement of the jellyfish. Taking a single image is, therefore, ideal, however GSD is a crucial factor that allows the identification of smaller jellyfishes and more accurate measurements (since each pixel is a given distance). To alleviate this issue, drones used for jellyfish research should have the highest camera resolution possible that allows smaller ground sampling distance (higher-resolution images) at higher altitudes. This approach allows the user to accurately measure the abundance and biomass of jellyfish aggregations while sampling larger areas. If a jellyfish aggregation is larger than the area that can be sampled by a single image at that altitude, the pilot may decide to take the image at a higher altitude at the cost of ground sampling distance, or use a drone model with a higher-resolution camera. If neither solution is feasible, then multiple successive images must be taken. In this case, a drone that has a camera with a mechanical shutter is crucial to eliminate rolling shutter effects that would warp images and lead to inaccurate orthomosaics [27] during rapid drone movement. Rolling shutter effects are one of the reasons using video for these purposes is not desirable, as the drone is moving at relatively high speed and video is especially prone to this effect.

Automated image analysis requires a clear contrast or colour between the jellyfish and the surrounding water, otherwise shapes that are not jellyfish may be mistaken for such by automated processing. Ideally, this means selecting an environment with a homogeneous water colour and clearly visible jellyfish (usually dark blue/green/brown with white jellyfish). The automated approach is unlikely to be accurate in very clear waters where the substrate is visible. In those cases, manual image processing may be necessary.

2.1.2. Drone Selection for Jellyfish Research

All jellyfish studies have been conducted from boats [25,26], with a multirotor rather than fixed-wing aircraft. In rolling seas, drones with large landing legs (i.e., the DJI Phantom) are preferable

for safety reasons. For deployments that can be from shore and/or for very large jellyfish aggregations, a fixed-wing drone may be appropriate.

### 2.1.3. Unmanned Aerial Vehicle (UAV) Flight Plan

Flight altitude should be primarily selected on the approximate size range of jellyfish and the ground-sampling distance required. A single image at that altitude should be captured if it covers the whole area of interest. If not, an interval image capture setting should be used that takes images every ~2 s during flight in a straight direction that will cover most of the aggregation in question. Flight speeds must be fast enough that jellyfish do not have time to move significantly between successive image captures, approximately 10–20 km.h$^{-1}$. U-shaped search patterns typically used for mapping will not work due to this effect as this increases the time between overlapping images. Because the whole process is quick, few flights are necessary to sample the selected aggregations outside those used to identify any aggregations.

### 2.1.4. Field Validation

Since jellyfish may be unevenly distributed across depth ranges, conducting a parallel field validation that relates 2D jellyfish densities and size with volume is necessary to accurately assess total biomass. This requires some depth-based sampling to identify typical depth distributions for the study species and the area. Previous studies have either conducted field assessments in parallel [26] or relied on previous studies in the same field site [25,28].

### 2.1.5. Image Processing

The relatively simple shape of jellyfish facilitates automated image processing, which increases accuracy while reducing processing time for examining large aggregations. In both Raoult and Gaston [25] and Schaub, et al. [26] images required digital manipulation before analyses. This was primarily to increase the contrast between jellyfish for automated image analysis software, with the aim of reducing images to two-tone white and black with only jellyfish visible. Images can then be stitched together using conventional orthomosaic software, which will provide information on true ground sampling distance and allow measurements to be taken.

### 2.1.6. Example Protocol for Studying Jellyfish with Drones

Increases in jellyfish abundance were identified during routine activities in Smiths Lake, New South Wales, Australia (152.519° E 32.393° S). Using a 4.8m vessel, the pilot along with the drone (DJI Phantom 3 Pro) were taken to an area within the lake where a jellyfish aggregation was identified. The drone was launched from the vessel and piloted in an approximate U-shaped search pattern to locate the aggregation and the likely orientation of the transect (since the aggregation was too large to capture in one image). The optimal altitude for flight was identified by choosing a height where individual jellyfish were still clearly visible from the tablet screen. For the targeted relatively large jellyfish species (*Catostylus mosaicus)* that grows to a 30 cm bell width and drone configuration (12 MP camera) this was 70 m for a ground sampling distance of 5.7 cm. Images were manually captured at intervals to obtain 80% overlap between images. The drone was flown in a straight line across the identified aggregation at ~10km/h for approximately 500 m, or 12 images in total. Images were then colour-corrected and processed into an orthomosaic.

## 2.2. Protocols for the Use of Drones for Shark Research

### 2.2.1. Shark Traits

Published research on sharks using drone-based methods has become commonplace in the scientific literature over the last five years. These studies have largely focused on detection and abundance estimates [19,29], and tracking and behavioural observations [25,30–35]. Drones typically

survey comparatively smaller spatial scales compared to manned aircraft [36,37], and while sharks may be rare in abundance [38], they are often easy to detect from drones if conditions are ideal [39,40]. Observations are generally confined to small embayments, shallow island reefs and nearshore coastal strips or islands. The error incurred in drone-based detections of sharks relative to true densities largely depends on water depth, sea conditions, water clarity, and the animal's position in the water column [39–42]. However, unlike surveying air-breathing marine species, many shark species are far less prone to swimming at the surface and spend considerably more time at depth [43]. The clear exception to this might be whale sharks and basking sharks [41,44]. In general, one would expect low detection reliability for sharks, particularly if they are prone to predominantly not be near the surface and the seabed cannot be distinguished [39,40,42]. Furthermore, it can be difficult to differentiate many species of sharks from the air, such as between various whaler sharks (*Carcharhinid* spp.), particularly if they are relatively deep for the given water clarity. However, the rapid advancements in artificial intelligence might significantly improve the identification of morphologically similar species of sharks from drones in the near future [45].

### 2.2.2. Impact of Drones on Sharks

Unlike many air-breathing marine species, sharks do not appear very prone to disturbance effects from small drones, as the frequency and loudness of the noise travels poorly through water [46]. This allows drone sampling to be highly useful for providing quantifiable aerial observations of sharks from relatively low altitudes without significant disturbance effects [29,31]. Drones have also proven useful for behavioural observations and fine-scale tracking of various shark species [30,31,42,47]. Typically, obtaining data on shark movement and behaviour has largely been conducted using boat-based observations [33,48,49], underwater video [50], or more commonly, satellite and/or acoustic tracking methods [51,52]. However, in comparison to satellite tracking methods, drones provide highly detailed information at much finer scales. When using drones for tracking or behavioural observations, the main limitations are flight time, aviation regulations of line-of-sight (or variations thereof, such as 'extended' line-of-sight), and water clarity and depth [36,39]. In addition, it is difficult to differentiate individual sharks from the air, so once a shark is lost from view, it is difficult to ascertain whether a subsequent sighting is the same individual or not, which can complicate analysis [48]. Due to the limitations of drone-based tracking, sharks are generally followed for up to tens of minutes [30,47]. In contrast, satellite tagging can track known individuals for long periods of time (up to years). However, a location fix is reliant on the shark occasionally swimming at the surface, which results in indications of very broadscale movement patterns. Therefore, the suitability of drones for tracking will depend on the environment they occur in, and precision and scale required, and is complimentary rather than a replacement for other tracking methods.

### 2.2.3. Drone Selection for Shark Research

Multirotor drones have proven useful for obtaining imagery from which real-world measurements can be extracted. In common off-the-shelf multirotor drones, the combination of the inertia management unit, barometer and GPS/GLONASS positioning system, allows vertical and horizontal positioning precision of ±0.5 m and ±1.5 m, respectively [12]. Therefore, with the gimbal orientated in nadir (pointed directly down) and the drone positioned directly above, the position of the shark can be obtained as well as photogrammetric measurements [30]. Regular calibrations of the inertia management unit and the gimbal are highly recommended to maintain the highest achievable measurement precision. The error margin in altitude can be greatly reduced (<1 cm) with RTK positioning; however, this adds considerable cost and requires at least one fixed-base unit. Measurements are also more accurate when the animal is at the surface, positioned across the centre of the sensor to minimise lens distortion, and with consideration of the compromise between pixel resolution for the given size of the shark and drone altitude. The lower the drone, the greater the potential relative error margin of the recorded altitude of the drone (since the error is constant), but the higher the resolution for pinpointing the

extremities of the shark. For example, Raoult, et al. [47] tracked <1.25 m (total length) Epaulette sharks (*Hemiscyllium ocellatum*) and >1.25 m reef sharks (*Carcharhinus melanopterus*, *Negaprion acutidens*) in a shallow reef lagoon using a <2 kg drone with a 94° field-of-view camera at 2–5 m altitude; whereas Colefax, et al. [30] tracked 2–4 m white sharks (*Carcharodon carcharias*) along coastal exposed beaches using a similar a <2 kg drone with a 94° field-of-view camera, but tracked at 20–25 m to better observe behavioural interactions and record length estimates.

The effects of water surface distortion on photogrammetric measurement precision is typically considered negligible when conducted on whales, as these large animals are regularly at the surface [53]. In comparison, many species of sharks are much smaller and are most often submerged. This issue creates a greater source of precision error when obtaining measurements than the potential error from the drone's altitude. This is because of the effects of sea-surface distortion on the submerged target, which is also residing at an unknown depth. Sea-surface distortion can be largely compensated for by capturing the animal at a high framerate, or high-resolution video, and scrolling through to find still frames with minimal distortion to extract multiple measurements [30]. However, compensating for the depth of the animal from the air may be challenging, with any associated offset term likely having to be approximated from an estimated depth. Due to this, morphometric measurements or relative size are best estimated when sharks are near the surface in calm and clear water.

### 2.2.4. Example Protocol for Studying Sharks with Drones

During each flight drones were flown at 8 m/s (30 km/h) and at an altitude of 60 m with the camera at nadir (pitched 90 degrees to face downwards). This gave a search width of approximately 110 m. Flights paths were generally designed as a transect, with the inside edge of the viewable area lining up with the 'backline' of the surf break. As the position of the surf break was observed to change significantly due to tide and weather variables, so the flights were made with manual control (as opposed to automated flight paths). Each flight path extended up to 1 km either side of the ground control station, covering a 4 km flight circuit. During poor visibility conditions, which restricted line-of-sight distances regarding the drone, transect length occasionally required shortening to maintain Civil Aviation Safety Authority (CASA) protocols. The drone cameras were equipped with circular polarising filters (2-stop neutral density circular polarising filters) to minimise sea surface reflection (glare). Additionally, sun hoods were used to shade iPads to minimise screen glare. Video recording options were set to a video size of 4k (3840 × 2160) at 25 fps.

When a shark was sighted, video was immediately recorded and then the drone lowered to a height of 15 m where a size estimate could be obtained, and then the shark was tracked until a battery change was necessary or the shark moved on to deeper water or beyond line-of-sight limitations. The position of the drone was maintained with the shark in the centre of the screen with the heading of the shark aligning with the forward aspect of the drone, as much as practical. Shark tracking was done at an appropriate height to suit conditions and the situation. As a pre-determined size reference, when the drone is at 15 m, the outer white ring of the record button on the iPad represents 2 m. If the shark was sighted within 100 m of water users, or 200 m of water users and swimming towards them, then pilots were to activate evacuation procedures. Where public safety becomes a concern, pilots could break shark tracking procedures to change orientation as required or leave the shark to alert water users. Sirens attached to the drones proved to be the most effective evacuation alert tool.

### 2.3. Protocols for the Study of Marine Reptiles

### 2.3.1. Traits of Marine Reptiles

Marine reptiles include the seven species of sea turtles, sea snakes and kraits and marine iguanas, with saltwater crocodiles also seasonally frequenting coastal habitats. To date, only the coastal habitats of sea turtles and saltwater crocodiles have been surveyed with drones.

Sea turtles are long-lived vertebrates that spend most of their lives in the marine environment, with immature and mature individuals being widely dispersed across oceanic and coastal developmental, foraging and wintering habitats, often located hundreds to thousands of kilometres from breeding grounds (for overview, see Rees, et al. [54]). Consequently, it is very difficult to observe and count individuals in a given population and obtain information on density, distribution and dynamics to facilitate holistic conservation management [55]. Existing data continue to be primarily based on the counts of adult females or their nests when they emerge to nest on beaches (for details, see Pfaller, et al. [56]). As such, for many regions of the world, we still do not know how many adult males and immature turtles there are in sea turtle populations, the distribution and use of foraging, developmental and wintering habitats, or how different species and populations mix in these areas [54,55]. In part, these issues are associated with the limitations of standard marine surveys (i.e., underwater, boat-based, land-viewing platforms), typically requiring high effort, cost, favourable (safe) sea-state and weather conditions, difficulty in accessing deeper or more offshore areas, and ease of detecting sufficient animals underwater to be representative at a population level [57]. Thus, drones have the potential to help fill this knowledge gap through identifying and monitoring habitats used by the different life stages both spatially and temporally [2,54]. In fact, the superior ability of drones to detect sea turtles in the marine environment over traditional boat-based or underwater-based count approaches has been confirmed [58]. In parallel, the various payloads on drones provide an opportunity to evaluate how sea turtles optimise their use of the various habitats they frequent at different life-stages in relation to real-time environmental/climatic factors they frequent at different life-stages, and at the population level, which could provide insights on their ability to adjust to our fast-changing climate [59].

Saltwater crocodiles are primarily distributed in south east Asia (from India to the north coast of Australia), and tend to inhabit freshwater swamps during the tropical wet season, where they mate and lay eggs, migrating up to 600 km to estuarine and coastal areas in the dry season [60]. Consequently, their coastal habitat use directly overlaps with the period when sea turtles breed and nest on beaches. Crocodiles use the beaches, surf zones [61], and can even spend several weeks at a time at sea [60]. Thus, there is huge potential to use drones to explore how the habitat use of marine reptiles overlaps with other marine animals and habitats. Regarding saltwater crocodiles, drone surveys have focused on evaluating the extent to which they are disturbed in coastal habitat [61] and their scavenging activity of a whale carcass in the marine environment [62].

Within the last 10 years, the feasibility of drones to explore almost every life-history stage of all seven sea turtle species has been demonstrated, including hatchlings (offspring at emergence from nests and initial dispersal and predation risk from the coast), immature turtles and adult males and females in the waters of foraging and breeding habitats (including evaluations of feeding, mating and cleaning behaviours, distribution and relative sex ratios), as well as adult females emerging on beaches to nest [14,16,17,20,57,58,63,64]. Furthermore, there have been studies characterizing nesting beaches and detection as part of broader marine-megafauna surveys [14,19,21,65].

### 2.3.2. Drone Selection for Marine Reptiles

Primarily, commercially available multirotor craft were used (n = 10 for sea turtles and n = 2 for saltwater crocodiles, i.e., DJI Phantom series) over fixed-wing craft (n = 3). In two of the studies, the fixed-wing craft were military grade craft, not typically available to researchers, with the capacity to operate for >1 h and tens to hundreds of kilometres outside of the visual line of sight [21,65]. The published studies used a variety of altitudes, ranging from 5 m to 122 m. To monitor hatchlings on the beach and in nearshore waters, Tapilatu, et al. [64] flew at an altitude of 5 m. To record beach characteristics (photogrammetry), whereas Varela, et al. [66] used an altitude of 30 m.

Bevan, et al. [17], Bevan, et al. [20] demonstrated that immature and adult sea turtles (green, hawksbill, flatbacks) showed no evasive behaviours (e.g., rapid diving) to drone altitudes of 10 and 20–30 m in shallow coastal waters, respectively, while adult flatbacks on beaches (crawling and

nesting) were not disturbed by drones flying forward or stationary at 10 m altitude. However, [61] demonstrated that the behaviours of saltwater crocodiles were noticeably disturbed by drones flown at altitudes below 50 m, when basking on the beach, in the surf and when actively swimming in the water. This disturbance was attributed to the auditory capacity of crocodiles being concentrated at low frequencies and their visual acuity [61]. Other studies of various crocodilian species in freshwater habitats flew at altitudes of 75 to 300 m [67–69]. Saltwater crocodiles are much larger than sea turtles (adults are around 3–6 m versus 1–2 m, respectively), facilitating detection at greater altitudes. For sea turtles, at 50–60 m altitude, it is possible to distinguish species, infer the sex of adults (based on males typically having tails that protrude from the carapace) and estimate body length (to distinguish size/age classes) [58].

For the commercially available drones, surveys were conducted daily or at 2–3 week intervals, depending on the objective. Sets of surveys were conducted over areas extending along 3 km to 55 km of shore and up to 1 km offshore at 100 m to 500 m intervals. Drones were flown parallel to shore in all surveys (except when departing/returning to the controller). Foraging turtles and adult females at breeding areas tend to show minimal movement (i.e., generally remaining within the same 100 m field of view when hovering), whereas males in breeding areas tend to swim parallel to shore seeking females for mating opportunities in breeding areas [58]; thus, flying parallel to shore reduces the chance of repeat sightings across transect lines. Most existing studies flew the drones along pre-programmed routes, while a few used a combination of forward and stationary manual flight. Stationary flight allowed the paths and speed of movement of multiple animals to be documented at once.

### 2.3.3. Methodological Difficulties

All 15 of the currently published studies on sea turtles and salt-water crocodiles were conducted in the daytime and in coastal areas. One study limited surveys to seabed depths of 6 m [19], while another validated that detecting turtles at greater than 7 m seabed depth under optimal conditions is very difficult [58]. This means that surveys at greater distances offshore (and greater seabed depths) must integrate standard distance sampling techniques to estimate the abundance of populations [70].

The limited battery time of commercial unmanned aerial vehicles (UAVs) is less of an issue when flying transects compared to stationary surveys of focal individuals to obtain information on behaviour. For instance, Bevan, et al. [17] were not able to document the full duration of mating, as males can remain mounted on female turtles for 1 h or more, and it was not possible to relocate the same individuals between flights. As sea turtles have the capacity to remain submerged for 30 min to 1 h between breathing bouts, this could hinder monitoring other behaviours too. For some activities, this could be overcome by targeting focal sites (e.g., fish cleaning stations; [58]), rather than focal individuals, or by using at least two drones that fly in succession when the battery of one drone is depleted as per Colefax, et al. [30].

Another key challenge is extracting information on individual turtles, particularly when population size is large and surveys are conducted regularly. Neural networks to automate this process are already being explored for sea turtles [71], 16.3% precision and 76.5% recall at one site. However, this issue is more pronounced in non-uniform habitats, such as seagrass beds and reefs (typical foraging habitats), compared to uniform habitats, such as sandbanks [72]. This issue might ultimately lead to drone survey bias to certain habitat types used by sea turtles.

### 2.3.4. Example Protocol for Studying Marine Reptiles with Drones

Variation in the relative abundance of adult male and female loggerhead sea turtles (*Caretta caretta*) was detected in the breeding area of Laganas Bay, Zakynthos Island, Greece (37°43′ N, 20°52′ E). From set launch points, the drone (DJI Phantom 3 Pro) was flown at a range of altitudes from 30 m to 100 m along line transects at 50 m, 150 m, 250 m and 350 m from shore (max. seabed depth approx. 4 m), allowing 50 m and 100 m fields of view. Sixty metres was ultimately considered to be optimal with the 100 m spacing as it provided full coverage, turtles (approx. 0.7 to 1 m body length)

and sex (tails extending beyond body in males) could be distinguished, with no overlap in footage (as females were primarily resting and males were primarily moving parallel to shore in search of females). Drone speeds of 12 m/s were considered sufficiently fast enough to prevent repeat sightings of individuals across transects (with mean swim speeds of 4 m/min at the breeding area). The drone was operated in continuous flight mode and the video footage was viewed at the processing stage only. Footage was manually processed by two independent observers, and still images were used to resolve any discrepancies in detecting individuals or discriminating sex.

*2.4. Protocols for the Study of Marine Birds*

2.4.1. Traits of Marine Birds

Determining the likely interaction between birds and drones and subsequent protocols is a natural curiosity, since both can share the same airspace. Drones have offered a new set of methods to study bird populations, particularly in the context of estimating population size and other metrics related to breeding habits and reproductive success. Unfortunately, it is often during breeding periods that these bird populations are at most risk to disturbance events. The natural predators of many bird species are larger birds, so drones have the potential to evoke disturbance responses. Yet, there is a long history of documenting the disturbance effects due to proximity to human observers, so drones still offer the potential for reduced disturbance relative to other survey methods.

Seabirds are generally long-lived with high reproductive costs, so are thought to be more likely to skip or abandon breeding in unfavourable conditions or when highly disturbed [73]. Yet individual heterogeneity can be large [74] and sensitivity to drones varies widely, even among similar seabird species [75,76]. Thus, it is of primary interest to determine species- and status-specific responses to drones to balance information collection with disturbance, and whether generalised protocols for monitoring seabirds are appropriate.

2.4.2. Impacts of Drones on Marine Birds

The interaction between birds and drones has garnered significant interest in the literature, and many studies exist that are dedicated to the interaction between seabirds and drones. There is very limited work that directly monitors physiological responses (an exception is Weimerskirch, et al. [76]) due the difficulty of deploying a monitoring apparatus in birds, thus studies typically report on behavioural responses. The findings of Weimerskirch, et al. [76] are a good overall representation of the literature in the context of guidelines to plan drone-based monitoring of seabirds; most seabird species show no detectable reactions when a drone is 50 m or further away

Larger colonies seem to be less prone to disturbance compared to small, more sparsely nested colonies. Gulls show limited reactions when drones are greater than 15–40 m away [77,78] and return to the nest quickly [75]. Penguin responses vary among species but typically show limited reactions beyond 10 20 m [79,80]. Petrel species are among the more sensitive species with notable reactions at around 50 m [76]. Albatrosses and Skuas show limited responses beyond 25 m [76]. Ibis species are tolerant to drones down to about 15 m [81]. Redshanks will flush at a flying height of 10 m [82]. Many other species (shearwaters, terns, scoters, cormorants, divers) have been surveyed successfully at higher altitudes (50–100 m+) but specific threshold altitudes are not well studied [83–85].

Overall, there is a large observed variance both within and among species, thus a general precautionary approach of beginning at a 50–100 m distance to the target species seems appropriate. Approach distances and flying altitudes can be lowered as required to fulfil the monitoring requirements, until an unacceptable level of disturbance is observed.

2.4.3. Selection of Drone for Marine Bird Research

Typically, drone selection for monitoring marine birds is initially guided by environmental considerations; namely take-off and landing space and prevailing wind conditions. For example,

in flooded estuarine environments it may be impossible to find enough space for a fixed-wing landing zone, thus a multirotor would be required, or a landing system that involves a waterproof landing (e.g., net/line catch, parachute). Many coastal locations also have strong prevailing winds (>30 km/h) for large proportions of the year, therefore a drone platform able to cope with high wind speeds would be required. The general rule of flying at 50–100 m altitude seems to be equally applicable to multirotor and fixed-wing platforms, but there is anecdotal evidence that birds of prey are more likely to be threatened by fixed-wing platforms [86]. The next consideration is the extent of the target, and whether it is a large colony or a small group/few individuals. Large colonies (extent in km) are best surveyed with a fixed-wing platforms, but can still be successfully surveyed with multirotor platforms if fixed-wing access is not possible [81]. Similarly, individuals spread over very large extents, where their location is unknown, would also be most appropriately surveyed via a fixed-wing platform. In general, when there are many targets with known locations, spread across large extents, a fixed-wing drone represents the most efficient option. In the case of birds, this issue usually applies to nesting locations for individuals or small groups. A final specific example for marine birds is cliff nesting birds; due to their aspect, fixed-wing platforms are rarely able to capture imagery at an appropriate incident angle.

### 2.4.4. Potential Hazards and Methodological Difficulties

There are many potential hazards when monitoring bird species with drones. The most obvious is the ability of bird species to physically interact or attack the drone, though this seems to be a relatively rare occurrence when operating drones in a sensible and considerate manner. Potential hazards in the context of breeding include adults leaving nests unattended or abandoning nests altogether, which is quite species specific. A precautionary approach may be to fly drones during post-hatching or post-fledging periods when practical or possible.

The biggest methodological limitation for monitoring seabirds with drones is typically the ability to effectively distinguish individuals in the imagery, whether this is due to incident angles required for image acquisition [75], camouflaged or hidden nests [83], large numbers and highly mobile individuals [81], or very sparse individuals [76].

### 2.4.5. Example Protocols for Studying Marine Birds with Drones

Seabirds are diverse and cover a very wide range of life histories and ecologies (e.g., shore nesting, cliff nesting, pelagic, estuarine, colonial, open nesting), all of which will have varying study site characteristics and required protocols. Here we have covered a range of considerations relating to a selection of species that cover that main variation in seabirds.

Most studies using drones for monitoring seabirds have historically, and continue to be, focused on estimating population sizes, often with the intention to further infer reproductive metrics like breeding status or breeding success [76]. Some studies replicate a plot-based type of survey by taking and analysing individual photos at some predetermined set of locations and heights, with either manual or automated counting techniques (e.g., Hodgson, et al. [62]). With the popularisation of photogrammetry techniques, image acquisition now mostly involves capturing many photos across an entire colony or habitat extent (e.g., multiple overpasses, grid patterns), with the intention of creating an image mosaic to conduct an exhaustive count. These counts can also be via manual counting (e.g., Ratcliffe, et al. [77]) or some type of automated approach (e.g., Rush, et al. [78], Groom, et al. [79]). Most studies have found drone-based counting methods are as good or better than field-based methods, with a range of resource and practicality trade-offs. Regardless of image capture requirements, all drone-based studies of marine birds should use the precautionary approach, that is: (1) take-off location beyond the expected disturbance distance, (2) testing disturbance distances before undertaking the flight missions and (3) continuously monitoring the shared airspace for early detection and avoidance of potentially harmful interactions.

### 2.5. Protocols for the Study of Pinnipeds

#### 2.5.1. Pinniped Traits

Global pinniped populations have dwindled over the past centuries due to anthropogenic interference. Historically, many species were hunted for food and other products which drastically lowered population numbers, while more recently they face challenges from commercial fishing, pollution, and climate change [87–90]. Pinnipeds are an apex predator, so monitoring their populations is a good indicator of ecosystem health and productivity [91]. As such, pinniped colonies have been monitored for decades to assess the long-term changes in their distribution and abundance. Pinnipeds are often monitored at haul-out sites using traditional methods such as observer counts, capture-mark-recapture, and manned aerial surveys. However, these methods have their limitations. Pinnipeds are often densely packed at haul-out sites, making observer counts difficult [92], while capture-mark-recapture is both labour intensive and very disruptive to the population [93], and manned aerial surveys are expensive and often low resolution [94]. Pinnipeds are very exposed when hauled out on land, and are easily spooked by the presence of observers, causing them to flush into the water. This can cause stampedes, injuries, disruption to breeding, and even result in the death of individuals [95,96].

Small drones offer a new approach for monitoring pinniped populations, removing many of the difficulties of previous methods. Drones can launch from hundreds of meters away from the haul-out, greatly lowering the likelihood of observers being detected. Pinnipeds are also much less wary of drones compared to ground-based observers as they do not have any aerial predators. In addition to reducing disturbance, drones also offer several advantages for data capture and research outcomes. The imagery captured creates a permanent record of the data, which reduces observer bias [97], it is faster than observer counts, so there is less time for individuals to move, and correctly captured data can also be used to create spatially accurate orthomosaics, digital surface models (DSMs), and 3-dimensional models for animal health metrics and condition assessment [98,99]. UAV pup counts also detected up to 32% more pups than observer counts [100]. While drone imagery may not detect individuals under ledges or overhangs, this still suggests that drones capture more of the colony than a ground-based observer count.

#### 2.5.2. Impact of Drones on Pinnipeds

While drones offer many advantages over traditional methods, they also introduce new possible disturbances. There have been several publications examining the disturbance drones pose to pinniped populations [100–105], and the two main disturbances identified are the noise and silhouette of the drone disturbing the colony. Early publications also highlighted the possibility of odour disturbing the colony, primarily from the fuel, but the transition from combustible fuels to electric propulsion for drones for both noise and weight reductions removed this variable. Below, we provide a list of recommendations for drone remote data collection of pinniped colonies.

#### 2.5.3. Drone Selection for Pinniped Research

Both multirotor and fixed-wing airframes have been used for pinniped population monitoring [104–106]. Fixed-wing airframes are best suited to population counts, and only recommended if large geographic areas require monitoring. Under most conditions, multirotor airframes are the best option for pinniped monitoring.

We recommend a sub-2 kg take-off weight for the airframe. This size of airframe greatly reduces the noise generated as there is less weight to lift, and reduces the silhouette of the airframe in the sky. Hexacopter configurations can provide motor redundancy compared to the quadcopter configuration, but they also tend to be heavier. As such, our recommendations are based on drone take-off weight instead of configuration. This weight also limits the use of cameras to those configured for drones instead of handheld 40+ MP digital single lens reflex (DSLR) cameras, but airframes which can lift such

cameras are at least two times larger, and therefore need to be higher above the colony, negating the advantage of the high-megapixel cameras.

Flying height corresponds to both the noise and silhouette of the drone, and is dependent on the weight and size of the drone. Current publications vary on "acceptable" flying heights of between 23 m (75 feet) [103] and 50 m (164 feet) [107]. There have been rare observed disturbance responses when flying at a range of over 200 m away [104], or as high as 60 m (195 feet) above a colony [108], so flying height is subject to the disturbance response of the colony in question. Based on our research, we recommend a minimum height of 30 m (100 feet) for a sub-2 kg airframe above a pinniped colony. Flying at a minimum height of 30 m, the only response we have observed is the occasional pinniped looking up at the drone while still achieving sub-centimetre resolution in the imagery.

Flying method can affect the noise generated by the drone. While accelerating, the motors rotate faster, generating greater noise. Similarly, a sudden stop also causes acceleration of some motors and greater noise. As such, we recommend using an automated flight path of survey lines with continuous motion. This method generates a much more "constant" sound while over the colony as opposed to manual flight, or pausing for each image. Pausing may also not be necessary with a drone equipped with a manual shutter and moving at slower (~10 km/h) speeds. The survey lines should go beyond the colony so that stopping and turning the airframe does not occur over the colony itself. We also recommend launching from at least 100 m from the colony to reduce noise impacts.

Flight design is dictated by research outcomes rather than minimising disturbance. Depending on the capture method, drone imagery can be used for population counts, body measurements, and even condition assessments. We recommend a flight design which allows for all three research outcomes: a cross-hatch or lawnmower-pattern flight design with 70% front overlap and 70% side overlap. While this level of overlap is high, it accounts for variation in the height of the terrain. The cross-hatch flight design does double the time in the air, and can also result in greater movement of individuals between images, but it provides a greater number of images for triangulation, and therefore greater accuracy in measurements.

The ability to collect sub-centimetre resolution imagery across an entire colony provides a great opportunity to generate metrics which can be used to assess body condition [99]. This requires ground control points to geometrically correct the mosaicked imagery and remove distortions that could potentially bias measures extracted. A minimum of 10 GCPs should be placed both in and around the colony, and marked with a survey-grade GPS. These can be permanent markers, placed outside of the breeding season or when the pinnipeds are not present, and used for an entire observing season or more, but they must be affixed so they cannot be moved by the pinnipeds when they haul-out. The GCPs prevent warping in the data modelling and increase both the relative and absolute accuracies of the resultant models. In instances where GCPs are not possible, the data can still be used for counts, but not body measurements. The availability of real-time kinetic GPS (RTK) and associated base stations can mitigate the use of GCPs to some degree, but should not be relied on when GCPs are easy to incorporate.

### 2.5.4. Example Protocol for Studying Pinnipeds with Drones

We used drones to collect population data of an Australian fur seal (*Arctocephalus pusillus doriferus*) colony on Kanowna Island, Wilsons Promontory, Australia. We used the DJI Phantom 4 Pro V2.0 quadcopter (Da-Jiang Innovation, Shenzhen, Guangdong, China), as it is a reliable and cost-effective option which is equipped with a 20 MP camera, weighs approximately 1.4 kg, and has "low noise" propellers. Before surveys, we placed 20 GCPs measuring 10 cm within the mapping region, and marked them with an EMLID Reach RS RTK GPS (Emlid, Saint Petersburg, Russia). We placed the GCPs outside of breeding time, and did so during the hottest part of the day when almost all the seals were in the water. We placed 20 GCPs to account for some being moved or obscured by seals during the remote data capture. The drone was launched from approximately 100 m away from the mapping area, downwind of the colony, and on high terrain to allow for clear vision of the drone. Flight plans

were conducted in Pix4DCapture autonomous flight-planning software. Due to the height difference between the launch area and the colony, the flying height at the launch point was set at 20 m with 55% front and side overlap, which equated to at least 30m above the colony, with approximately 70% front and side overlap. Two flights were undertaken to cover the entire colony. The data were processed using structure from motion techniques generating an orthomosaic and DSMs with an average resolution of 0.92 cm/pixel as well as point clouds and triangular meshes with a mean root mean square error of 0.015 cm generated from 18 GCPs. These data were used to assess population numbers, male seal territories, and seal condition with a range of body metrics extracted from a georeferenced orthomosaic image and DSM imagery [99].

### 2.6. Protocols for the Study of Sirenians

#### 2.6.1. Traits of Sirenians

Sirenians include four living species: the dugong (*Dugong dugon*), the Amazonian manatee (*Trichechus inunguis*), the African manatee (*T. senegalensis*) and the American manatee (*T. manatus*). The American manatee encompasses two subspecies: the Florida manatee (*T. m. latirostris*) and the Antillean manatee (*T. m. manatus*). While dugongs exclusively inhabit marine areas, manatees can be found in riverine, estuarine, and marine environments, with exception of Amazonian manatees, which occurs exclusively in freshwater habitats of the Amazon basin. Sirenians are slow moving animals, rarely presenting superficial behaviors and normally exposing only part of the muzzle when breathing [109]. With few exceptions including large aggregations around resources, sirenians are primarily solitary or form small transient groups [109,110]. Therefore, despite their relatively large body size, sirenians are often inconspicuous and can be difficult to detect in areas where their densities are low and/or where aggregations do not occur.

Globally, sirenian populations face several threats from hunting and boat collisions, to habitat loss and modification. Aerial manned surveys are used to estimate dugong and manatee densities [111–113], particularly in shallow coastal areas with good water transparency. In riverine and lentic habitats consisted in brownish waters, side-scan sonars have been employed as a suitable method to detect manatees (e.g., [114,115]). Over the last decade, however, drones have emerged as an affordable, non-invasive, and effective tool to monitor sirenians, with a great potential to elucidate several aspects of their ecology, biology, and behavior.

#### 2.6.2. Impact of Drones on Sirenians

Drone flight altitude seems to be a key factor regarding sirenians responses to drone. Although low altitudes are desirable to improve ground-sampling distance, the trade-off is that increasing the proximity could also elicit undesirable behavioral reactions of the target animal. Ramos, et al. [116] assessed the behavioral responses of Antillean manatees to small multirotor drones (*DJI Phantom*), detecting manatee responses on 24% of all sightings. Manatees responded at drones flying from 6 to 52 m altitude, generally by fleeing the area and evading the drone when pursued. Most response events occurred below 30 m flight altitude, and direct approaches (vertical descents) were more likely to cause responses than either stable hovering or horizontal follows. Even when no behavioral reactions are observed, manatees could experience changes in respiratory rate and general activity levels. The experience and personality of individual manatees may also influence their responses to drones. Avoiding direct approaches and selecting the highest flight altitude feasible for acquiring quality data should prevent disturbance [116]. The continuous improvement of drone technology, including reduced noise and better camera resolution and inbuilt zoom is expected to decrease the potential bias associated with sirenians' reaction to drones.

### 2.6.3. Drone Selection for Sirenian Research

Both multirotor and fixed-wing drones can be used for the purpose of detecting sirenians (e.g., [15,117]). On the other hand, multirotor drones can maintain stationary flight, allowing the capture of high-quality images and videos of surfacing marine mammals [114], which can be useful for studies based on photogrammetry and photo-identification. For low-budget studies, multirotor drones are appropriate for sirenian research. Concurrent drone flights in an area can be used to assess all these aspects of sirenian ecology, and modifications of flight protocols to optimize various objectives can be made on the fly (Figure 3).

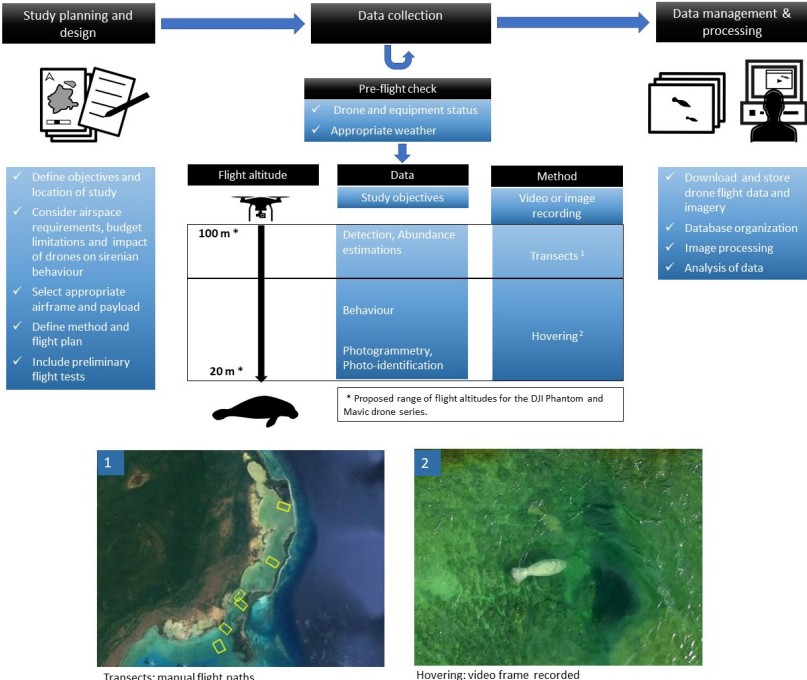

**Figure 3.** Example operational protocol flowchart for studying sirenians using drones. Of note, the species, airframe, payload and study objectives are used to define flight parameters. Yellow squares indicate flight paths used to search for sirenians. Image (2) credit: adapted from Eric Ramos

### 2.6.4. Detection and Density Estimations

Compared to manned flights, drones are more suited for local-scale surveys [36]. To detect sirenians, the flights can be programmed to follow parallel line transects or grids of georeferenced images can be automatically captured covering the target area (e.g., [15,118]), or can be manually controlled with a video transmitted live to the ground operator (e.g., [116,119]). Manual flights are especially useful in areas of low densities because the animal position can be identified in the field, allowing the recording of additional data, such as imagery suitable for further studies and behavior. In areas where sirenians aggregate, the hovering mode of multirotor drones is a recommended approach for detection and monitoring abundance (e.g., [120]). When using a transect method during flights, the camera should point directly downwards so that the width of the image or the video represent the transect strip width. Higher flight altitudes allow for a wider transect strip width but reduces image resolution. The optimal flight altitude for sirenian detection depends largely on the drone payload and associated ground sampling distance. For example, Hodgson, et al. [15] used a Nikon D90 12 megapixel digital SLR camera mounted in a fixed-wing *ScanEagle* drone, allowing dugongs to be detectable on images from altitudes of 305 m, resulting in a strip width of 144 m and ground sampling distance of approximately 3 cm/pix. Landeo-Yauri [121] used an off-the-shelf drone (DJI Phantom 3 Advanced) to detect manatees on aerial 2.7 K resolution videos from a flight altitude of 100 m, resulting in a strip

width of 154 m and ground sampling distance of approximately 4 cm/pix on images (3840 × 2160 pix) extracted from videos. It is prudent to conduct test flights when using a new drone or imaging system to minimize issues in the field and maintain consistency in data. Post-flight processing includes manual revision of visual data [15,121], although algorithms for automatic detection on aerial images from drones are under development [122], such as the Dugong detector software, which includes machine learning and allows mapping of dugongs and seagrass distributions [123]. Note that these flight altitudes are near or above airspace restrictions for drones in some areas, and appropriate permits may be necessary.

### 2.6.5. Photoidentification and Photogrammetry

Sirenians often bear scars, mutilations, and congenital deformities on their bodies that are stable enough to re-identify individuals over time [124]. Since drones obtain overhead views, they are ideal for capturing images on the dorsum of sirenians, which is the part of the body where the most conspicuous boat-related injuries are typically found [125]. Landeo-Yauri, et al. [117] proposed a protocol for manatee photo-ID using multirotor off-the shelve drones, extracting suitable images from aerial videos, selecting them according to image quality and distinctiveness of manatee features, and compiling a catalog of distinctive individuals. The most adequate images for photo-ID were obtained below 30 m flight altitude using DJI Phantom 3 and Phantom 4 drones. Photogrammetry methods for manatees are still under development but are proposed to estimate metric equivalences from pixel measurements as with baleen whales [126]. This would allow the remote assessment of body condition status, growth monitoring, and many other applications.

### 2.6.6. Behavioral Studies

The behavior and interactions of sirenians can be monitored using a multirotor to hover over known aggregation areas. For example, Landeo-Yauri, et al. [127] collected information on interactions between tourism boats and manatees using a drone hovering over known hotspots for manatees. Ramos, et al. [116] used hovering and horizontal drone flights to assess manatee reactions to the drone, tracking the movement of the manatees by matching the information of the GPS track from drone flights, aerial videos, and orthomosaic maps of manatee habitats. Being permanent records, aerial videos allow for all individuals previously registered to be evaluated. Moreover, objects of known size recorded on video can be used to estimate sizes, distances, and speeds relevant to behavior.

### 2.6.7. Capturing and Health Monitoring

Free-ranging manatees and dugongs are regularly captured in several areas globally for health assessment [128] or rescue purposes. American manatees are typically caught in nets deployed from a specialized boat [128], while the "rodeo" technique [129] has been successfully implemented for dugongs in Australia [130]. Drones recently proved to be a key tool during this process, and their applications range from rapid and efficient detection of the target animal and assessing the effectiveness of the procedure, until the analysis of potential challenges of the activity. Criteria such as the spatial location of animals, habitat features, net position, and vessel speed and direction can be verified in real time and transmitted, via radio to the crew on board [131]. When equipped with a infrared sensor, drones are also a promissory tool for non-invasive clinical diagnosis of captive and wild sirenians, allowing the healing process of external injuries to be monitored [131].

### 2.6.8. Methodological Difficulties

Regarding imagery for photo-ID purposes, the representative features and scars can be distorted or masked due to ripples, light diffusion on water, and turbidity [117]. It is also challenging to obtain manatee images depicting the whole body in straight position and near surface, most suitable for photogrammetry purposes. In these cases, the use of aerial videos allows for selection of the frames when feature and animal position are adequate.

The use of drones for the monitoring and conservation of sirenians has many potential applications yet to be explored, including habitat mapping [131], risk assessment (boat and human disturbance, habitat loss) and monitoring of rehabilitated individuals. As a complementary tool, drones can add interesting information to telemetry and tracking programs aiming to study movements, site fidelity and habitat use [132]. Sirenians are mostly distributed in developing countries, where their monitoring and research is hampered by economic constraints. With the improvement of drone technology, paired with lower costs of production, it is expected that more sirenian researchers and stakeholders will benefit from the use of this tool within the next years.

*2.7. Protocols for the Study of Odontocetes*

2.7.1. Traits of Dolphins

The use of drones as a safer and cost-effective alternative for collecting data on cetaceans is still limited to few specific areas of research. Drones have proved to be an invaluable asset to study changes in body condition, energetic costs, health and habitat use in various Mysticetes [13,133–136]. In contrast, the application of drones for studying odontocetesis still growing primarily due to the lack of data on the impacts of drones on the behaviour of the target species. The unpredictable surfacing patterns and behaviour render odontocetes a difficult target to follow, particularly in deep or turbid water even for experienced pilots. Among odontocetes, to the authors' knowledge, drones have only been implemented to study a few species of dolphins. Example applications have included identifying dolphin presence and habitat use [137–142].

2.7.2. Drone Selection for Dolphin Research

Two main types of drones have been used to study free-ranging dolphins; (1) existing off-the-shelf drone technology (e.g., DJI multirotors and (2) specifically designed, custom-built drone technology [142]. Off-the-shelf drones have been the type most frequently used to study dolphins. They are easy to use and more appropriate for less experienced pilots but their application in research is typically limited to observational-based studies. In contrast, customised drones allow for more dedicated research to be conducted but requires more specialised piloting experience. For example, custom appendages have been implemented on multirotor drones to allow the collection of dolphin microbiome to study dolphin health [141,142].

2.7.3. Behavioural Impacts of Drones on Dolphins

Dolphins critically rely on sound, hearing and vision to complete their normal daily activities including foraging, socialising and threat avoidance. Both visual stimuli such as the shadow produced by a drone, and the acoustic footprint generated from propellers, have the potential to disrupt dolphin behaviour [143] with repercussions on potential biases in data. To date, the behavioural responses of dolphins when approached by a drone remains poorly investigated, limited to a single species (Bottlenose dolphins, *Tursiops* sp.) and small drones of <2.5 kg.

Christiansen, et al. [46] undertook a detailed investigation of the ability of marine mammals to hear drone-produced noise underwater and concluded that while dolphins near the surface may hear the drone approaching, the propeller noise is likely to be masked by ambient noise. Although the underwater noise disturbance caused by drones flying at low altitude is expected to be small, there is growing scientific evidence reporting significant behavioural reactions of bottlenose dolphins to the presence of drones [141,143,144]. All available evidence suggests that when a small drone is flown at an altitude between 10–30 m above bottlenose dolphins, short-term behavioural responses occur [141,143,144]. Behavioural reactions were generally of short duration and largely limited to individual responses such as deep diving, turning toward the drone, side rolling, change in swimming direction and tail slap. These responses varied depending on group size and behaviour.

These studies had several limitations; Fettermann, et al. (2019) was the only study with an explicit experimental design to assess disturbance, but analysis was limited to resting behaviour and the sample size was small. Ramos, et al. (2018) and Gill et al. (2020) used opportunistic data to infer behavioural changes as an indication of drone disturbance and were not designed to quantify the responses of animals at different flight altitudes, and thus results could be biased.

Raudino, et al. [142] were the first to collect a dolphin respiratory blow via a drone and reported similar behavioural response to a drone in Australian humpback dolphins (*Sousa sahulensis*) when sampling for microbiome. However, the authors were unable to determine whether aggressive behaviours documented in socializing dolphins occurred in response to the drone. Strong behavioural reactions of Australian humpback dolphin and Australian snubfin dolphin (*Orcaella hensohni*) to the presence of a drone have been also observed by co-authors Cagnazzi D. and Colefax A., who noticed a small >2 kg drone (DJI Phantom 4 Pro) can trigger a visible behavioural response from ~50 m. Furthermore, the proximity for triggering a behavioural response is likely influenced by pod size, behavioural state, and wind velocity. This was observed during an on-going study on dolphin-UAVs interaction in Queensland (Cagnazzi and Colefax pers. comm.).

Overall, our understanding of the impact of drones on dolphin behaviour remains scarce and limited to a single species. Whether the observed reactions are a response to the visual or the acoustic stimuli remain unclear and to date no studies have attempted to assess the variation in acoustic behaviour of dolphins with the presence of a drone. Despite the limitations, all current studies have highlighted the complexity of assessing disturbance responses in dolphins in the presence of a drone. This confirms the urgent need for more ad hoc studies to assess the impact of drones on dolphins before general flying protocols are produced to conduct safe and responsible drone operations near dolphins but also minimise the introduction of bias in data collection.

Considering the likely behavioural differences among species, populations and habitats, as well as differences in noise levels between drone models, an assessment of disturbance before initiating each study is strongly advised. When assessing disturbance, a range of variables that may affect behavioural reactions, such as species, life stage, behaviour, group size, habitat characteristics, drone model and the duration and frequency of the exposure should be considered. Masking factors, such as water visibility, sea state, wind speed, habitat geomorphology and external disturbance sources need to be quantified and included in the assessment process.

For research purposes, until behavioural responses are better understood, we suggest following a precautionary approach and fly below 30 m only if necessary for <2 kg drones, and higher for larger models. Social structure or behavioural studies may not require a close approach, but rather, a full view of the entire group, and can, therefore, be conducted from higher altitudes with current payload technology. Advances in camera resolution mean that videos can be digitally cropped with retaining adequate resolution for interpretation and analysis, allowing images to be collected at a safer distance. For health assessment studies that require the collection of biological samples and morphometric measurements, drones need to be flown closer to the target individual. In these cases we suggest following generally recognised principles to minimise impacts on dolphins such as avoiding flying over mother and calf pairs, minimising the flight time over the same group, avoiding close approaches in socialising groups, and interrupt data collection when there is evidence of strong behavioural responses.

Knowledge gained from disturbance assessments will provide invaluable guidance to update existing regulations for drone operations targeting marine mammals, including odontocetes, which are either lacking or derived from existing regulations for manned aircraft. Under the Australian National Guidelines for Whale and Dolphin Watching, for example, drones are considered as manned aircraft and the minimum approaching distance is 300 m above whale and dolphins, where drones would have difficulty obtaining clear footage of dolphins. However, under Australian CASA regulations drones cannot be flown higher than 120 m above ground level. This creates a situation where most drone operations observing odontocete behaviours are not permissible under normal flight rules.

Under the Australian National Guidelines for Whale and Dolphin Watching, for example, drones are considered as manned aircraft and the minimum approaching distance is 500 m above whale and dolphins, where drones would have difficulty obtaining clear footage of dolphins. However, under CASA regulations drones cannot be flown higher than 120 m above ground level. This creates a situation where most drone operations observing odontocete behaviours are not permissible under normal flight rules.

*2.8. Protocols for the Study of Baleen Whales*

2.8.1. Traits of Baleen Whales

Baleen whales are large aquatic mammals (Order CETARTIODACTYLA, Suborder MYSTICETI; Committee on Taxonomy, 2020) that range from 6 m (20 ft) to the 31m (102 ft) in length, making them difficult to study using typical methods applied to small mammals that can be captured and handled. There are currently 14 recognized species (Committee on Taxonomy, 2020) that all share a common anatomical feature—baleen plates hanging from their upper jaw—that allow them to efficiently filter zooplankton and small fishes from the water and achieve such large sizes (e.g., Goldbogen, et al. [144]). Baleen whales tend to hear best at lower frequencies (10 Hz to 10 kHz, see Popper, et al. [145]). Baleen whales are found in both coastal and pelagic environments and many species make significant annual migrations from high-latitude feeding grounds to low-latitude breeding grounds (e.g., Robbins, et al. [146]). In these locations they often congregate in high densities [147], facilitating the study of their biology and ecology [148].

Drones are increasingly applied to the study of baleen whales and provide opportunities for new observations that are generally not available through traditional non-invasive means [149]. In general, these applications can be separated into two categories: 1. Studies focused on estimating abundance or density of baleen whales in a particular geographic region (hereafter referred to as population sampling) and, 2. Studies examining individual animal behaviour [150], morphology [151], pigmentation patterns, and a range of health and welfare metrics such as scarring and entanglement rates [152], respiratory microbiomes [13,153], body size and condition [126], and even internal body temperature [154] (hereafter referred to as individual sampling). Distinct operational protocols for these two general application types are required, as different aircraft, sensors, control systems and flight plans are needed to safely and efficiently collect robust data. Also, within each of these categories there may be specific requirements based on species, sampling needs, and local environmental conditions.

2.8.2. Reducing Disturbance

At present, there is very little evidence that drone flights, whether for population or individual sampling, disrupts the behaviour of baleen whales [155]. Furthermore, it remains unclear which acoustic or visual stimulus would drive these behaviours. At least one study indicates that flying directly toward the rostrum of the whale may elicit a stronger response [156], yet most studies that employ drones to study baleen whales indicate that whales rarely if ever react to the presence of a drone [13,155]. Experimental work indicates that very little sound produced by typical drones is transmitted through the surface of the ocean [46], and at present there is no experimental work detailing responses generated by visual stimulus from either the drone or its shadow projected onto the water. It should also be noted that both visual and acoustic stimuli produced by a drone would be minor in comparison to those presented by occupied aircraft, or nearby vessels, and any potential disturbance from drones should be considered in that context.

2.8.3. Drone Selection for Baleen Whales

*Population Sampling:* a high-endurance aircraft is optimal for population assessments in baleen whales, which are often found across broad regions, even when congregating on feeding and breeding grounds. The most successful programs employ military-grade fixed-wing systems that require

significant operational overhead in comparison to typical occupied aircraft surveys (e.g., [157,158]). These systems, however, can survey for baleen whales (and other marine mammals, see section on sirenians) over extensive areas, and can produce abundance or density estimates with similar quality of that produced by occupied aircraft surveys [157], although some caveats remain (e.g., [158,159]). Consumer off-the-shelf multirotors may be applied to conduct population sampling in smaller areas, with a concomitant limitation on inference regarding the broader context of the target species abundance/density. There are a growing number of small transitional aircraft (vertical take off and landing, VTOL) that combine the launch/recovery simplicity of multirotor aircraft with the endurance of fixed wing systems. These platforms are leading to new applications for surveying relatively large areas for marine megafauna presence. In terms of payloads, population sampling for baleen whales appears to be best accomplished with a medium to high resolution electro-optical (EO) red, green, blue (RGB) camera, often in the form of a DSLR camera (see Angliss, et al. [158]) or a digital mirrorless camera for smaller aircraft. In some cases, small consumer drones have integrated sensor packages (e.g., DJI Phantom series drones) or modular sensors that are specifically designed for the platform (e.g., senseFly Sensor Optimized for Drone Applications, or SODA). While drone-based surveys for many mammalian species on land are often conducted using thermal infrared sensors (e.g., Seymour, et al. [160]), this technique is not recommended for the sole sensor for drone-based baleen whale surveys. While some portion of a baleen whale's body may be detectable above the surface with thermal infrared (IR) sensors (as are their blows; [154]), animals that are submerged are generally lost from view in thermal IR imagery, yet remain available in the visible part of the electromagnetic spectrum and still detectable via an EO RGB sensor. Combinations of EO RGB and thermal IR may be helpful under some circumstances.

### 2.8.4. Individual Sampling

Drone-based sampling of individual baleen whales is best accomplished with multirotor platforms, which provide the immediacy and operational flexibility required to access these animals in a timely manner. Multirotor aircraft are usually small and portable, allowing them to be deployed rapidly from small boats in response to an often unpredictable presence near the surface of these animals that only surface for short periods before diving for upwards of 15 min [161]. Multirotors enable at-sea operations that are more proximate to study animals in general. Both custom and consumer off-the-shelf multirotors are used to study morphometrics and kinematics [153], pigmentation/scarring patterns [152] and the body size and condition of baleen whales [126]. In situations where highly accurate measurements are essential, researchers should not rely on the onboard barometric altimeter data for photogrammetric analysis. Barometric altimeters are subject to considerable variability both within and between flights, and are likely the largest source of error that must be accounted for in any measurement programs for baleen whales as they have been in sharks [30]. To overcome this limitation, researchers should adopt the use of a laser or radar altimeter on their drone, which will provide accurate (+/−3 cm) measurements of altitude that are critical for robust photogrammetric studies (e.g., Dawson, et al. [9]). In some cases, data can be collected with a physical scale (e.g., a boat of known length) within the camera field of view to support scaling image pixels to real-world units [135], but this requires increased and likely unnecessary proximity to study animals which may influence their behaviour, and incurs operational costs that can be avoided with the use of a quality altimeter. Multirotors are also essential for biological sampling of respiratory microbiomes of baleen whales, and these systems are usually custom built for the application (e.g., [13]), or modified commercially available aircraft fitted with a mechanical sampling apparatus designed to capture respiratory vapours from whale blow [153]. These systems are often navigated via a first-person view (FPV) camera on the forward-facing portion of the drone airframe to line up the aircraft and sync sampling runs with periodic blows of the animal. In terms of sensors, individual sampling of baleen whales for external characteristics and body measurements requires a high-resolution EO RGB sensor with a global (mechanical) shutter and a low distortion lens. As the pixels of each image will ultimately be

converted to real-world values, it is optimal if individual pixel dimensions remain uniform across the image. For many lower cost (and wide-angle) lenses, there can be considerable distortion across the frame, with the largest distortions at the corners and edges of the image. Low distortion lenses help alleviate these problems, but often at a financial cost. In some cases, specific lens-camera combinations can be paired to further reduce distortion through physical and electronic corrections, and through software workflows that correct camera distortions [162]. Distortion can be alleviated using software post-flight, but will not produce results as good as with a low-distortion lens.

2.8.5. Flight Planning, Control and Other Operational Considerations

Flight planning is a key consideration for both population and individual sampling of baleen whales using drones. For drone-based population sampling, flight plans for baleen whales often follow similar principles to those applied to standard strip or distance sampling protocols, where a series of parallel lines are programmed perpendicular to cross environmental gradients that may structure organismal density. These surveys are usually flown autonomously via a ground control station to maintain standard speed and direction, both of which are difficult for manual piloting over the areas required for these types of surveys. In contrast to many drone-based surveys for marine organisms, surveys for baleen whales are not likely to generate orthomosaics of the regions produced through structure-from-motion photogrammetric processing, largely because keypoint matching across large areas of changing ocean surface is not feasible. Instead, researchers should consider flights as sampling transects where video or still imagery are collected and then processed to estimate the number of animals detected in the area covered by the imagery. For still image sampling, flight plans that collect imagery that reduces the potential for double counting images are important (e.g., Sykora-Bodie, et al. [63]). Depending on the location, logistical constraints, and expected density of study animals, the spacing between lines can be optimized with implications for the amount and quality of data collected. Occupied aircraft surveys for baleen whales often have different detection modes, as human observers in aircraft can be cued by a variety of whale behaviours such as animals lunging at the surface, breaches and blows at distance (e.g., Ferguson, et al. [159]) whereas most drone-based systems simply record the surface of the ocean for later assessment. This can result in very different detection rates and survey coverage between unoccupied and occupied aircraft surveys. These factors should be considered when adopting drones for population assessment surveys for baleen whales.

In contrast, drone-based sampling of individual baleen whales is usually not conducted using a pre-programmed flight path. In almost all cases the drone is controlled manually for most of each flight, as the remote pilot continually positions the aircraft above or near the animal in question according to sampling requirements. After launch by hand or via a vessel-based launching platform, the aircraft will be flown rapidly to the location of a whale surfacing, as the animals are available at the surface for restricted periods of time. In many cases remote pilots will not fly on the first detection of a baleen whale. Instead, the drone will be launched after the behaviour of the animal has been assessed (e.g., dive times, length of surface interval, number of breaths per surfacing, speed and direction of travel). This pre-flight assessment can provide the remote pilot with the opportunity to coordinate drone launches to best capture the time that the whale is at the surface. This practice is especially useful for many short endurance multirotors that are employed for whale photogrammetry. In some cases, the safe flight times (e.g., allowing extra for time for recovery or poor battery performance in colder locations) of these drones is short—in the order of 12–15 min, which in some cases is comparable to the dive times of many baleen whales. Once over the whale, still or video imagery of the animals is collected through as many surfacings as possible. High frame rate image sequences are more useful than single image captures, as they provide more flexibility for image analysis later, allowing for the selection of the most suitable image when the animal is flat and close/at the surface. Researchers should make every attempt to connect drone sampling with boat-based photographs, behavioural observations and biopsy sampling, as these connections will help analysts identify individual whales and maximize the utility

of drone-based sampling data. This often requires well-synced boat and drone-based photography and capturing greater areas of the animal body in boat-based imagery to aid in matching to the overhead image perspective from the drone. Furthermore, it is often essential to sync image capture times with GPS time (and ideally with timestamps from the laser altimeter if one is being employed) to match the appropriate altitude with each photo or frame grab. This provides for situations where rapid changes in altitude may occur, through purposeful piloting or turbulence. Integrated camera/drone systems often do this automatically, but in cases where this is not possible, pilots should obtain a photographic timestamp to capture any offsets between camera time and details of the flight recorded by the flight controller such as altitude, pitch and roll.

### 2.8.6. Example Protocol for Studying Whales with Drones

Prior to studying whales with drones, all efforts must be made to establish a safe and secure platform for drone launch/retrieval (if captured by hand) or a secure take off/landing position from land or onboard a vessel. Once this is established, a search via dedicated observers (if available) will take place to locate whales in the area. This usually involves identifying pod/s location, species and behaviour e.g., direction of movement (if any) and surface intervals. Once a suitable individual or pod has been identified as appropriate for sampling e.g., relaxed and consistent in their movements (course maintained and predictable surface intervals preferable), sample collection or whale measurements can progress to the next stage. For whale blow collection, this involves placing a sterile petri dish in the appropriate location/s on the custom drone. Drone operators will remain on standby and ensure safety around the drone prior to take off, while dedicated observers monitor whale behaviour. If a whale/s are moving consistently, observers may be able to provide the drone pilot with an estimated count down until the whale/s resurface, during which time the drone may be launched prior to the surfacing or launched as the first surfacing happens. Once in the air, the drone pilot may be able to locate the whale/s position without assistance or communicate with observers to assist with positioning near the whale/s. If in a pod, the drone operator may be able to choose an individual to focus on based on natural markings/colourations/scars (if observable) from the first-person view of the drone. Once within close range of a whale (<10 metres), the drone may be able to hover adjacent to their position and wait for a surfacing. Sun glare and other weather conditions e.g., wind, may make this difficult at times. In some cases, drone operators may be able to see whales underwater prior to surfacing from the air. Once the chosen whale surfaces, the drone is deliberately flown through the densest part of the blow. Depending on the size of blow, the drone can be flown within three to 10 metres of a whale. Once a sample is collected, the drone is immediately flown to a higher elevation (>30 m) and flown safely back to land or the vessel, where a safe landing can be conducted until the next flight.

## 3. Synopsis

### 3.1. Impacts of Drones on Marine Animals

Across the phyla examined here, there were clear differences in reactions of marine animals to the presence of drones with some showing physiological or behavioural signs of distress and others exhibiting no measurable reaction. The impacts of drones on animal welfare are driven by many factors that include drone shape or flight patterns which may be similar to those of potential predators (e.g., for seabirds) or the noise frequencies that overlap with those used in intra-specific communication (e.g., odontocetes) or auditory capacity (e.g., saltwater crocodiles). Conversely, animals with simplistic sensory systems (e.g., cnidarians) or that have no aerial predators (e.g., sea turtles) have little to no reaction to the presence of drones. In this context, and given that no marine animal research occurs in isolation and is likely to encounter non-target animals from other taxa that may be impacted by the use of drones in research, we recommend drone pilots choose flight plans that account for the more vulnerable phyla that may be encountered rather than only considering the targeted species. Across all drone-sensitive species, altitudes greater than 30 m rarely resulted in any adverse reaction beyond the

acknowledgement of the presence of the drone (e.g., looking in the direction of). We suggest, therefore, that where possible drone research on marine animals is conducted at these altitudes as a minimum, provided target animals are still detectable. In addition, we suggest research on the possible vectors of disturbance from drones (audible, visual) are further examined. Identifying noise or size thresholds that are appropriate for studying a given taxon would allow more unified and accurate responses to mitigate disturbance as new drone airframes become available.

### 3.2. Use of Drones to Study Fishes

While fishes are a ubiquitous organism in marine ecosystems, we were surprised to find few if any studies that used drones to study fishes. Typically, drones were used indirectly to assess habitat associated with fishes (e.g., [163–165]) or to assess fishery impact [166,167]. We suggest this is an area of research that could be greatly expanded given the success of drones with similar organisms such as sharks. Aspects of social interactions and movement could be examined as they have been in sharks, and species that occasionally surface such as Sunfish (*Mola mola*) might also be studied in ways like baleen whales. This may include non-destructive fisheries stock assessment techniques for schooling species potentially visible from the surface or document migrations such as those from anadromous fish where water clarity allows.

### 3.3. Lack of Use of Vertical Take Off and Landing (VTOL) Aircraft for Marine Research

Multirotors and fixed-wing drones are used in most of the studies examined here, and typically the two platforms have trade-offs with flight time, areal coverage, and ability to take off from small areas. VTOLs (also known as transitional aircraft), which aim to combine the benefits of fixed-wing aircraft (longer range and flight time) with the multirotor-like ability of taking off from restricted landing fields, have not been widely adopted in this field. Since VTOLs combine aspects of multirotors and fixed-wing aircraft, they are typically more complex than either of those two airframes, and as such are generally more expensive than more conventional options (e.g., $16,000 for a DeltaQuad Pro). Cost difference may partly explain the lack of adoption of VTOLs, yet these prices are not significantly different from more pro-sumer grade multirotor or fixed-wing aircraft (e.g., Phantom 4 Pro RTK, $12,000). It is possible that their perceived complexity and cost may deter users, even though they offer similar autonomous capabilities as both other platforms. We suggest that researchers looking for the flight characteristics of fixed-wing drones but who are deterred by the required take-off and landing areas seriously consider adopting VTOLs in their research.

### 3.4. Legislation

Drone research operates in a legislative environment that is rapidly shifting and variable across nations that can impact the uptake for marine research. Drone use is are often regulated under civil aviation authorities with laws in place to minimise the potential impact to the public, manned aircraft, and aviation space. Privacy and security issues are often discussed in association with drone use, and a good understanding of local legislative requirements is necessary to conduct drone research in a way that will not hinder future projects [168]. Inappropriate flight practices from scientists are likely to have long-term negative impacts on the field, so we underline the need for scientists using drones to follow local best-practice and legislation as much as possible.

In Australia, new laws regarding the operation of drones and certification requirements are making it easier for researchers to explore the application of these sensor platforms for research and monitoring programs [99,169]. This includes the operation of drones up to 2 kg to be flown without a CASA (Civil Aviation Safety Authority) certification required by the operator, called the Excluded Category (CASR, 1998). This still requires registration of the operator, understanding of the safety rules, and the registration of airframes from September 2020. This opens up the potential of establishing citizen science programs using drones where systematic surveys can be conducted under stringent protocols to ensure the welfare of animals and humans. Citizen science projects using drones may

provide the opportunity for great spatial and temporal coverage than can be achieved by science teams as well as the benefits associated with empowering knowledge in communities. Drone-based citizen science has started organically with, for example, recordings of informative marine animal behaviours or submission of shark sightings via apps. There are now bespoke drone-based citizen science programs such as the Victorian Coastal Monitoring Program, which has more than 120 citizen scientists engaged in flying drones to map and monitor coastal change [170]. We recommend in areas where drones are already being used recreationally (e.g., along beaches) that scientists explore the potential of citizen scientists.

Common drone airspace regulations (altitude <120 m, visual line of sight, buffer distance to aerodromes) can limit research methods, although airspace authorities are making permits possible to fly outside these limits (e.g., CASA in Australia). Operations of drones is also at the discretion of the landowner/ manager and some public lands for example may not allow operations under the Excluded Category within their managed land. In these cases, operations under a commercial Remote Operator's Certificate (ReOC) may be permitted. The cost of an Operator's Certificate may create barriers for uptake and entry into this research field, halting the potential for development of the application of drones in studying marine animals in some regions.

The majority of the research presented here has focused on operations in visual line of sight (usually <500 m from the pilot) as a result of legislation, however, drones have the capacity to operate at much greater distances up to 30km for some models. Flights at these distances are known as beyond visual line of sight (BVLOS or BLOS), and legislation is currently being tested in many areas including Australia (CASA) and Europe (EASA, European Union Aviation Safety Agency) to allow flight in these operational parameters. Easier approval for flights in these conditions could allow researchers to cover much larger areas and may by association encourage more common use of fixed-wing platforms.

### 3.5. Safety and Personal Protective Equipment for Use of Drones in Marine Research

Drones can cause life-changing [171] or lethal injuries [172] when they interact with people, with larger drones becoming increasingly dangerous. Large helicopter drones have led to instances of decapitation, and more conventional drones have led to loss of eyes [171,172], and researchers need to take the safety aspect of drone operation seriously. Even unpowered drones can be dangerous when wind causes blades to spin, as one author can attest. The use of personal protective equipment is, therefore, essential during drone operations. These effects are compounded during deployments that require the hand launch and recovery of multirotor drones. Those involved with releasing and capturing multirotor aircraft should wear protective gloves (thick leather, kevlar reinforcements) to reduce risk of injury from propellers, and safety glasses/face shields and hard hats should be worn to reduce risk of injuries to the head and face in the event of equipment failure or pilot error. The use of take-off pads on mobile substrates such as sand can also prevent accidents, as drones are liable to create low pressures that bring foreign objects towards the drone and can cause crashes. For the safety of the public, we also recommend deploying clear signs that inform others about the presence of drones. Drone batteries are currently dominated by lithium-polymer designs that can spontaneously combust if overheated, overcharged, or undercharged. As a precaution, we recommend all drone batteries be transported in individual fireproof or fire-resistant containers or bags during charging, storage and transport, and that researchers carry a fire extinguisher into the field where possible. This is because unlike other fires, lithium polymer fires are not extinguishable with water [173]. Visual inspection of batteries can also identify potentially damaged units, although onboard battery monitoring technology helps users identify bad batteries before they cause trouble.

### 4. Conclusions

The use of drones to study marine animals clearly offers the potential to study a broad variety of taxa. While the protocols presented here are comprehensive, this research field is still novel, and these protocols are likely to evolve quickly in the near term as regulations and drone technology make

new drone-based methods possible. We recommend all marine researchers that focus on animals that occupy water near the surface to consider possible applications of drones to improve knowledge and management, and the contents of this review will assist them in developing a consistent approach where they have little previous experience with drones.

**Author Contributions:** V.R., A.P.C., P.A.B. conceived the study. V.R. managed compilation of comments and drafting. All other authors contributed equally to this review. All authors have read and agreed to the published version of the manuscript.

**Funding:** This research received no external funding.

**Conflicts of Interest:** The authors declare no conflict of interest.

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
