# Peer review of "Operational Protocols for the Use of Drones in Marine Animal Research"

_drones, doi:10.3390/drones4040064_

Round 1

Reviewer 1 Report

General comments:

In summary, this study claims to (point 1) offer a review of the application of drones to study marine animals and (point 2) uses a number of so called “experts” to (point 3) outline the current best practice for drone operations in marine environments. While this idea is overall good, I feel that the manuscript fails on all three points and hence should not be published in its current state. I describe the shortcomings for each point below.

In regards to point 1, the authors does a pretty good job in summarizing a large amount of information and grouping them into different taxon and subjects. However, the structure of the manuscript is somewhat haphazard, with the number and content of each sub-section varying considerable. Also the amount of information provided for each sub-section varies considerably. For this reason it becomes very difficult to compare taxon and getting a good overview of the paper. I think the review part of the paper would benefit greatly from being re-structured into topics rather than taxon. That would also reduce overlap and repetition from one taxon to another. For example, when talking about challenges such as glare, water visibility, wind speeds etc., it would make more sense to discuss these topics across taxon in the same section, rather than spreading it out over the entire paper. The same goes for the choice of drones, protocols etc. I think there are some general patterns that could be drawn across taxon, but because of the structure of the paper into separate taxon, that pattern does not come across clearly. Finally, in regards to review, if this paper is intended to provide a comprehensive review of drone research on marine animals, then more studies on fish (not only sharks) should be provided. If there are none, then the authors should at least discuss why that might be, and if there is any potential for drone research into this taxon. Without it, a major component of the marine ecosystem is missing from the paper, which makes it far from comprehensive.

In the abstract the authors claim that “inconsistent methodological approaches have previously led to difficulties comparing studies”, however this issue is never really described in the actual paper. When has this proven difficult? Are there any examples? If not, is it really an issue to adapt methodologies to specific case studies? Or are there examples where consistent methodologies have allowed comparative studies across geographical areas or similar? If this is something that the authors wish to encourage, then arguments should be put forward to why this is important, what could be done etc.

I also find the paper currently written as a series of recommendations, without the authors providing alternative approaches and arguments to why these alternative approaches might not be as good as the one’s suggested by the authors.

In regards to points 2 and 3, the paper claims to “draw on experts with a wealth of practical experience with marine phyla and drones”. While I am not doubting that each author has some experience in drone based research on marine animals, calling them all “experts” is pretentious, given that some have published very few papers in this field of research. How were these “experts” chosen? Given the affiliations, which is very Australian focused, this author list seems rather to be the result of a number of local researchers coming together to discuss this topic, then inviting a few external experts, before attempting to write a paper to lead the way forward in this field. I don’t believe that that’s the proper way to go if you want to get an actual representation of what experts in the field believe to be the best way forward. I think a quantitative measure should have been used when selecting authors, based on the individuals research effort, number of species he or she has worked with, geographical areas, type of drones used, and (very importantly) his/her track record in the field of drone based marine research. Without this, this paper is just an opinion piece by a random group of researchers, and should be published as one. That a random number of researchers should draw future directions for drone based marine animal research (a huge field) is borderline offensive to the many researchers who have far more experience and expertise in this field and who have not been given an opportunity to express their thoughts here. Based on this, I feel this paper has no support for making recommendations on current best practises for drone operations in marine environment and should not be published as such. Instead the authors should rewrite it as an opinion piece and make this very clear to the reader both in the title and the abstract.

In general, I find the paper very lengthy, confusing and biased towards the opinions of the authors. The paper also lacks a proper synthesis of the various parts at the end, which makes it hard to draw any general conclusions. Finally the paper could benefit from a more structured (perhaps a figure) set of recommendations of what to do for various taxon, study aims etc. At the moment the recommendations are very vague.

Specific comments:

Section 2.1.1. Could video be an option, where researchers could extract several frames to measure jellyfish?

Line 192: How do you account for the depth of a shark when estimating the total length using drone photogrammetry?

Lines 208-209: Water surface distortion can be a real issue when measuring body morphometrics of animals below the surface.

Lines 212-215: It would have been nice to see how errors could be incorporated into the measurements. For example if you measure 10 images from the same shark, perhaps the variation in body morphometric measurements could use used to obtain an error estimate of how accurate the measurement is.

Fig. 2. This figure is not really needed I think as the various steps are common sense to anyone doing research. Of course you start by planning and designing the study, then collect data and then process it. This is hardly something new to wildlife researchers, and neither are the various bullet points below them.

Lines 681: The study of Ramos et al. (2018) suffers from several limitations, including a lack of adequate control and also pseudo-replication. It should not be cited as there being evidence of dolphins reacting to drones.

Lines 799-801: This is a strong statement which is citing an “in prep” article. That is not appropriate.

Author Response

Dear Editor,

Thank you for the opportunity to modify our manuscript for possible publication in Drones.  We thank the reviewers for their very helpful comments and the we believe the manuscript has been greatly improved. We have adopted the reviewer comments and made corresponding changes in the manuscript. Our responses to the reviewers, along with changes made, are in red font below.

Reviewer #1

General comments:

In summary, this study claims to (point 1) offer a review of the application of drones to study marine animals and (point 2) uses a number of so called “experts” to (point 3) outline the current best practice for drone operations in marine environments. While this idea is overall good, I feel that the manuscript fails on all three points and hence should not be published in its current state. I describe the shortcomings for each point below.

In regards to point 1, the authors does a pretty good job in summarizing a large amount of information and grouping them into different taxon and subjects. However, the structure of the manuscript is somewhat haphazard, with the number and content of each sub-section varying considerable. Also the amount of information provided for each sub-section varies considerably. For this reason it becomes very difficult to compare taxon and getting a good overview of the paper. I think the review part of the paper would benefit greatly from being re-structured into topics rather than taxon. That would also reduce overlap and repetition from one taxon to another. For example, when talking about challenges such as glare, water visibility, wind speeds etc., it would make more sense to discuss these topics across taxon in the same section, rather than spreading it out over the entire paper. The same goes for the choice of drones, protocols etc. I think there are some general patterns that could be drawn across taxon, but because of the structure of the paper into separate taxon, that pattern does not come across clearly.

This is a valid point and one we spent much time deciding on. While we agree that organizing our review by ‘problems’ rather than taxa could be interesting, we agreed the aim of this review was to help researchers interested in the use of drones in particular taxa. We therefore felt that talking about broad themes would introduce large quantities of information which readers may find irrelevant for their particular animal. In addition, we found this approach would still require significant separation of text for particular taxa that have very different drone-based approaches (like jellyfish). We do agree that some broader introductions on major topics are be relevant (as the other reviewers highlighted), and so have expanded on the overview to facilitate this.

Finally, in regards to review, if this paper is intended to provide a comprehensive review of drone research on marine animals, then more studies on fish (not only sharks) should be provided. If there are none, then the authors should at least discuss why that might be, and if there is any potential for drone research into this taxon. Without it, a major component of the marine ecosystem is missing from the paper, which makes it far from comprehensive.

The reviewer is correct to highlight that fishes are a major component of marine ecosystems, however, to our knowledge there are no studies using drones to study fishes. Drone use for this taxon focuses more on habitats or fisheries rather than the fishes themselves. We agree that this is a knowledge gap that should be examined using some of the protocols for similar taxa like elasmobranchs. We have highlighted this in the synopsis with a separate section addressing this as a gap that should be further explored. Please see section lines 1094 - 1103.

In the abstract the authors claim that “inconsistent methodological approaches have previously led to difficulties comparing studies”, however this issue is never really described in the actual paper. When has this proven difficult? Are there any examples? If not, is it really an issue to adapt methodologies to specific case studies? Or are there examples where consistent methodologies have allowed comparative studies across geographical areas or similar? If this is something that the authors wish to encourage, then arguments should be put forward to why this is important, what could be done etc.

I also find the paper currently written as a series of recommendations, without the authors providing alternative approaches and arguments to why these alternative approaches might not be as good as the one’s suggested by the authors.

We have changed the phrasing here to “could” lead to difficulties. This is a well-known issue in other fields, but the reviewer may be correct stating that this may be hard to objectively demonstrate because the field is so young. This paper reviews most of the literature published on each taxon examined, and builds off those to highlight the protocols that have worked more effectively than others given the authors’ experience working on the taxa. We also highlight idiosyncrasies of each species because of their size, behaviour, habitat use, and how this actually limits potential alternative approaches in the published literature.

 In regards to points 2 and 3, the paper claims to “draw on experts with a wealth of practical experience with marine phyla and drones”. While I am not doubting that each author has some experience in drone based research on marine animals, calling them all “experts” is pretentious, given that some have published very few papers in this field of research. How were these “experts” chosen? Given the affiliations, which is very Australian focused, this author list seems rather to be the result of a number of local researchers coming together to discuss this topic, then inviting a few external experts, before attempting to write a paper to lead the way forward in this field. I don’t believe that that’s the proper way to go if you want to get an actual representation of what experts in the field believe to be the best way forward. I think a quantitative measure should have been used when selecting authors, based on the individuals research effort, number of species he or she has worked with, geographical areas, type of drones used, and (very importantly) his/her track record in the field of drone based marine research. Without this, this paper is just an opinion piece by a random group of researchers, and should be published as one. That a random number of researchers should draw future directions for drone based marine animal research (a huge field) is borderline offensive to the many researchers who have far more experience and expertise in this field and who have not been given an opportunity to express their thoughts here. Based on this, I feel this paper has no support for making recommendations on current best practises for drone operations in marine environment and should not be published as such. Instead the authors should rewrite it as an opinion piece and make this very clear to the reader both in the title and the abstract.

We have changed “experts” to “researchers” to reflect the reviewer's comments. Please see changes line 31.

 In general, I find the paper very lengthy, confusing and biased towards the opinions of the authors. The paper also lacks a proper synthesis of the various parts at the end, which makes it hard to draw any general conclusions. Finally the paper could benefit from a more structured (perhaps a figure) set of recommendations of what to do for various taxon, study aims etc. At the moment the recommendations are very vague.

This comment is in disagreement with reviewers #2 and #3 who agreed the “paper describes a clear framework of contemporary UAV approaches”.

Specific comments:

Section 2.1.1. Could video be an option, where researchers could extract several frames to measure jellyfish?

This is a good question. Video is an option; however, video always comes at a trade-off of resolution as well as a rolling shutter effect, neither of which are ideal relative to images. For jellyfish, this is really critical as the GSD is a critical aspect of accurate measurements, more so than with larger species (e.g. sharks) where video frames can be appropriate. Similarly, video can be appropriate for sea turtles when conducting video transects (since orthomosaics are hard to produce in open water). We have added a comment on this topic lines 237 – 239.

 Line 192: How do you account for the depth of a shark when estimating the total length using drone photogrammetry?

This is a good question and addressed somewhat in the point below. Generally, measurements are only made when the animal is close to the surface. In addition, taking measurements from higher altitudes can limit the issues associated with measurements at depths, but without fluid lensing being more commonplace, being able to account for depth (since it is unknown) is not possible.

 Lines 208-209: Water surface distortion can be a real issue when measuring body morphometrics of animals below the surface.

This is a good point and is compounded by the point above (depth). Typically, distortion is less the shallower the animal will be. As we outline lines 348– 358, there are approaches that can be used to minimise this, such as carefully selecting frames with less visible distortion, and only measuring animals that are close to the surface where distortion is less likely to have an impact.

 Lines 212-215: It would have been nice to see how errors could be incorporated into the measurements. For example if you measure 10 images from the same shark, perhaps the variation in body morphometric measurements could use used to obtain an error estimate of how accurate the measurement is.

With respect, the accuracy of photogrammetric measurements has been covered in depth in other studies across a range of fields. Off-the-shelf drones flying at ~20m altitude will get measurement errors < 1% (relative to altitude), though the positional accuracy (e.g. in absolute GPS coordinates) can be less than that without GLONASS systems or GCPs. See: Cunliffe et al. (2016), Putch (2017), Hugenholtz et al. (2016), Uysal et al. (2015) as examples. These levels of accuracy are even greater than those that could be obtained manually with a carcass since they ignore confounding factors like body width that typically lengthen measurements.

 Fig. 2. This figure is not really needed I think as the various steps are common sense to anyone doing research. Of course you start by planning and designing the study, then collect data and then process it. This is hardly something new to wildlife researchers, and neither are the various bullet points below them.

The aim of this figure is not to outline study design, but example protocols for the respective study taxa. We believe that the protocol for flight altitudes, drone choice etc. will be novel to those interested in drone research. We also highlight the discrepancy between reviewer #1 and reviewer #3, who highlights that this figure was really helpful and would want to see more throughout the manuscript. 

 Lines 681: The study of Ramos et al. (2018) suffers from several limitations, including a lack of adequate control and also pseudo-replication. It should not be cited as there being evidence of dolphins reacting to drones.

Respectfully, we disagree that this study should not be cited. Expecting controlled conditions in the wild to study animal behaviour is not realistic, and is actually more relevant than in controlled conditions (e.g. dolphinariums) where sound reverberation is characteristically different from natural conditions (see Duncan et al. 2016, covered for fishes in Popper and Hawkins 2019). In the absence of alternative studies, the Ramos et al. 2018 study is one of the few to explicitly attempt to measure reactions of dolphins to drones and it should be cited.

Duncan, A.J., Lucke, K., Erbe, C. and McCauley, R.D., 2016, July. Issues associated with sound exposure experiments in tanks. In Proceedings of Meetings on Acoustics 4ENAL (Vol. 27, No. 1, p. 070008). Acoustical Society of America.

 Lines 799-801: This is a strong statement which is citing an “in prep” article. That is not appropriate.

We agree it is a strong statement and that supporting it with a text in prep may not be appropriate. We have provided another article by Colefax et al. As a citation that also supports this statement and is published.

Reviewer 2 Report

General comments

The manuscript describes an interesting synthesis of contemporary UAV approaches in marine animal scientific  research. It addresses an important topic and can be a useful contribution to the research community. The topic is very broad, therefore it is difficult to summarize everything in depth  on few pages because it is confronted with different scenarios, different regulatory systems and different technologies in continuous evolution. The paper also includes the safety of people and UAV devices and this is important because many people approach this technology in a very simplistic way. The paper  can be an interesting starting point for creating procedures and guidelines to standardize the collection of data in scientific research on marine animals.

Given a well-described general framework, I suggest only small remarks that may make some concepts clearer.

Specific comments

2.5.3. Drone selection for pinniped research

VTOL technology can also find application in other types of scientific research on marine animals (not just pinnipeds). Mentioning the VTOL technology is very interesting but it would be useful to explain (also in another paragraph): new perspectives on scientific research, reasons beacuse the use is not yet widely widespread, general advantages and disadvantages.

3.2  Current limitations of drone technology

Inside this paragraph it would be useful to have a more complete and global synthesis of the limits of this technology, inserting some indications already mentioned in the article: batteries in cold areas (see polar areas applications for example), take-off and landing in difficult situations (for example on a boat), etc.

The working environment is certainly not easy, so it would be interesting to have also informations about other problems encountered more frequently. For example, how marine aerosol affects the lifespan of drones (my frequent problem), how it is managed, how magnetic and electrical interference can make problems, etc.

3.3. Legislation

Given that the paper  describes the ordinary flight operations in "visual line of sight - VLOS" modality , that would be interesting to cite (or remarks) about new openings of some legislation towards operations in    Beyond Line of Sight BLOS  (see EASA) and what opportunities they could provide in scientific research.

3.4. Safety and personal protective equipment

It's very interesting include safety and prevention inside scientific works (often given as a secondary interest or not mentioned). If available, it would be interesting to mention the accidents  or near missing  (involving the drone and / or people) that have occurred more frequently in these scientific research activities. Or, if not available, mention the lack of these data which would be useful to collect to help the scientific community concerned.

Photogrammetry

Aerial photogrammetry is often used and quoted inside the paper. It would be interesting for the reader to clarify how to make orthomosaic with non-stationary scenes such as sea / lakes when they have continuous wave activity. It would be helpful to know operational limits and how to deal with these situations.

4. Conclusion

The paper describes a clear framework of contemporary UAV approaches in marine  animal scientific  research.
In the conclusions (or in a specific paragraph) it would be interesting to cite the greater technological needs for future  scientific research regarding study of marine animals by using UAV.
It would also be interesting to cite specific needs of the scientific community inside the regulatory frameworks regarding UAV.

Author Response

Dear Editor,

Thank you for the opportunity to modify our manuscript for possible publication in Drones.  We thank the reviewers for their very helpful comments and the we believe the manuscript has been greatly improved. We have adopted the reviewer comments and made corresponding changes in the manuscript. Our responses to the reviewers, along with changes made, are in red font below.

Reviewer #2

__________________________________________________________

General comments

The manuscript describes an interesting synthesis of contemporary UAV approaches in marine animal scientific  research. It addresses an important topic and can be a useful contribution to the research community. The topic is very broad, therefore it is difficult to summarize everything in depth  on few pages because it is confronted with different scenarios, different regulatory systems and different technologies in continuous evolution. The paper also includes the safety of people and UAV devices and this is important because many people approach this technology in a very simplistic way. The paper  can be an interesting starting point for creating procedures and guidelines to standardize the collection of data in scientific research on marine animals.

Given a well-described general framework, I suggest only small remarks that may make some concepts clearer.

We thank the reviewer for their positive feedback and their specific comments below.

 Specific comments

2.5.3. Drone selection for pinniped research

VTOL technology can also find application in other types of scientific research on marine animals (not just pinnipeds). Mentioning the VTOL technology is very interesting but it would be useful to explain (also in another paragraph): new perspectives on scientific research, reasons beacuse the use is not yet widely widespread, general advantages and disadvantages.

VTOLs are certainly an interesting technology and we have added a section in the synthesis to that effect. Please see new section lines 1105 – 1118.

 3.2  Current limitations of drone technology

Inside this paragraph it would be useful to have a more complete and global synthesis of the limits of this technology, inserting some indications already mentioned in the article: batteries in cold areas (see polar areas applications for example), take-off and landing in difficult situations (for example on a boat), etc.

The working environment is certainly not easy, so it would be interesting to have also informations about other problems encountered more frequently. For example, how marine aerosol affects the lifespan of drones (my frequent problem), how it is managed, how magnetic and electrical interference can make problems, etc.

The reviewer makes some excellent points and we have added information relative to battery environment limits as well as issues with working environments in the new overview section. Please see new section lines 145 – 199.

3.3. Legislation

Given that the paper  describes the ordinary flight operations in "visual line of sight - VLOS" modality , that would be interesting to cite (or remarks) about new openings of some legislation towards operations in    Beyond Line of Sight BLOS  (see EASA) and what opportunities they could provide in scientific research.

The reviewer makes a good point about BVLOS or BLOS, and we have highlighted that this is being explored by the legislation. Please see changes lines 1153 – 1159.

3.4. Safety and personal protective equipment

It's very interesting include safety and prevention inside scientific works (often given as a secondary interest or not mentioned). If available, it would be interesting to mention the accidents  or near missing  (involving the drone and / or people) that have occurred more frequently in these scientific research activities. Or, if not available, mention the lack of these data which would be useful to collect to help the scientific community concerned.

We thank the reviewer for their support. We have included what published research on drone injuries we could find, but have expanded on some details to highlight the threat, and highlight that these need to be more regularly documented to help identify common issues. Please see changes lines 1161 – 1169.

Photogrammetry

Aerial photogrammetry is often used and quoted inside the paper. It would be interesting for the reader to clarify how to make orthomosaic with non-stationary scenes such as sea / lakes when they have continuous wave activity. It would be helpful to know operational limits and how to deal with these situations.

We agree that a bit more clarification on photogrammetry was appropriate, and have introduced the topic more clearly in the overview. Typically, constructing orthomosaics in areas like the reviewer suggests is not possible due to low numbers of tie points, this is discussed somewhat in the baleen whale section. Please refer to new section lines 94 – 106.

  1. Conclusion

The paper describes a clear framework of contemporary UAV approaches in marine animal scientific research.
In the conclusions (or in a specific paragraph) it would be interesting to cite the greater technological needs for future  scientific research regarding study of marine animals by using UAV.
It would also be interesting to cite specific needs of the scientific community inside the regulatory frameworks regarding UAV.

These are good points and we have addressed them in the new overview, as well as in a lengthened legislation section that now discussed the potential uses of BVLOS for research, and a section that discusses the potential use of VTOLs for research.

Reviewer 3 Report

General Comments

The authors present a review of protocols for use of drones in marine animal research. Drone based sampling is increasing exponentially and we need papers like these to help standardize methods and prevent harmful or poor practices. The paper is generally well written and has lots of useful information. The structure of the paper could use some work. I suggest laying out some generalized definition and methods before going into specifics. This will eliminate some of the redundancy in the sections, particularly the selection section. Spend a little more time describing the sensor types and what their capabilities are. I would eliminate discussion of specific computer programs as they will likely not remain relevant in the future. Overall with some revision this would be a nice addition to the literature.

Before jumping in some definition of terms would be useful to those just getting started, like what a payload is, or what on the ground resolution means, what types of airframes are there etc., or cite other work that lays out more basic information about using drone systems and photogrammetry in field that have used drones longer (Broussard et al. 2018; Oniga et al. 2018; Harris et al. 2019; Manfreda et al. 2019). Rather than having individual sections on platform selection I think it would be better to lay out the differences, strengths, and weaknesses of each type of platform as it relates to the general SOP for each type of study.

Specific Comments

Line 45-47: Awkward phrasing consider revising.

Line 100: A section that defines terms like fixed wing and multirotor would be good for first time users.

Line 104: I don’t think this statement is true. There are fixed wing platforms that have every bit of the resolution as multirotor. I would say the consideration of the flight being done is more important for picking the type of platform than sensor type.

Line 185: No example protocol section?

Line 459: Here and in the section on birds, is there research about the noise levels in terms of decibels of sound that illicit responses from the animals and can you provide guidance on what sound levels the airframes emit?

Line 571: More figures like figure 2 would be great.

Line 651: Why not say cetaceans in the header to keep consistent with other sections.

Line 656: Suggest deleting internationally.

Line 897: I think this section could be moved up if you provide a more generalized approach and consideration and then specific taxa based suggestions.

Line 926: I don’t know how useful this figure is, as it can change with the sensor.

Broussard, Whitney III, Glenn M Suir, and Jenneke M Visser. 2018. Unmanned Aircraft Systems (UAS) and Satellite Imagery Collections in a Coastal Intermediate Marsh to Determine the Land-Water Interface, Vegetation Types, and Normalized Difference Vegetation Index (NDVI) Values. ERDC VICKSBURG United States.

Harris, J. Mason, James A. Nelson, Guillaume Rieucau, and Whitney P. Broussard III. 2019. Use of Drones in Fishery Science. Transactions of the American Fisheries Society 148: 687–697. https://doi.org/10.1002/tafs.10168.

Manfreda, Salvatore, Silvano Fortunato Dal Sasso, Alonso Pizarro, and Flavia Tauro. 2019. New Insights Offered by UAS for River Monitoring. Applications of Small Unmanned Aircraft Systems: Best Practices and Case Studies. CRC Press: 211.

Oniga, Valeria-Ersilia, Ana-Ioana Breaban, and Florian Statescu. 2018. Determining the optimum number of ground control points for obtaining high precision results based on UAS images. In , 2:352.

Author Response

Dear Editor,

Thank you for the opportunity to modify our manuscript for possible publication in Drones.  We thank the reviewers for their very helpful comments and the we believe the manuscript has been greatly improved. We have adopted the reviewer comments and made corresponding changes in the manuscript. Our responses to the reviewers, along with changes made, are in red font below.

Reviewer #3

General Comments

The authors present a review of protocols for use of drones in marine animal research. Drone based sampling is increasing exponentially and we need papers like these to help standardize methods and prevent harmful or poor practices. The paper is generally well written and has lots of useful information. The structure of the paper could use some work. I suggest laying out some generalized definition and methods before going into specifics. This will eliminate some of the redundancy in the sections, particularly the selection section. Spend a little more time describing the sensor types and what their capabilities are. I would eliminate discussion of specific computer programs as they will likely not remain relevant in the future. Overall with some revision this would be a nice addition to the literature.

We thank the reviewer for their positive comments and constructive feedback. We have removed most references to software except in the new section in the overview.

Before jumping in some definition of terms would be useful to those just getting started, like what a payload is, or what on the ground resolution means, what types of airframes are there etc., or cite other work that lays out more basic information about using drone systems and photogrammetry in field that have used drones longer (Broussard et al. 2018; Oniga et al. 2018; Harris et al. 2019; Manfreda et al. 2019). Rather than having individual sections on platform selection I think it would be better to lay out the differences, strengths, and weaknesses of each type of platform as it relates to the general SOP for each type of study.

We thank the reviewer for their feedback. We agree that a broader introduction that discusses both photogrammetry and various drone platforms and payloads is warranted and could reduce discussion in other sections. We have added these in the re-worked overview, and have integrated some of the suggested references where appropriate. Please see changes lines 63 – 205.

Specific Comments

 Line 45-47: Awkward phrasing consider revising.

We have revised this sentence for clarity.

Line 100: A section that defines terms like fixed wing and multirotor would be good for first time users.

See comment above and new section in the Overview section.

Line 104: I don’t think this statement is true. There are fixed wing platforms that have every bit of the resolution as multirotor. I would say the consideration of the flight being done is more important for picking the type of platform than sensor type.

We agree with the reviewer’s comment and have removed this statement.

Line 185: No example protocol section?

We have now included an example protocol section for every taxon except the sirenian section, which has the example protocol graphic instead. We have also not included one for working with folphins, since there is still high uncertainty about appropriate protocols for these animals.

Line 459: Here and in the section on birds, is there research about the noise levels in terms of decibels of sound that illicit responses from the animals and can you provide guidance on what sound levels the airframes emit?

This is an excellent point. In the synopsis on impacts of drones, we have highlighted that the mode of disturbance from drones should be further investigated, and these should be separated into audible vs visual stimuli. See changes lines 1088 - 1092.

Line 571: More figures like figure 2 would be great.

We appreciate that the reviewer found this figure helpful, however, this is at odds with reviewer 1 who did not like this figure, so we have not added additional figures. We have however expanded and produced dedicated sampling protocol example for each taxon.

Line 651: Why not say cetaceans in the header to keep consistent with other sections.

We have used ‘odontocete’ for consistency as suggested but have kept dolphin elsewhere in this section for ease of interpretation.

Line 656: Suggest deleting internationally.

We have deleted ‘internationally’.

Line 897: I think this section could be moved up if you provide a more generalized approach and consideration and then specific taxa based suggestions.

We agree that some of these sections would be more appropriate in the overview rather than the synopsis, and have moved sections relating to the payloads, and drone technology limitations into the overview. We have also added sections on photogrammetry, different airframe types, and the typical payloads available to most drones. Please refer to new sections lines 63 – 206.

Line 926: I don’t know how useful this figure is, as it can change with the sensor.

We initially had this figure with multiple camera resolutions but it quickly became too busy. To make it more broadly relevant, we have added in the caption the comment that higher or lower resolutions will change the intercept of that relationship and the GSD/area surveyed. Please see revised caption lines 121 – 124.

Round 2

Reviewer 1 Report

While the authors have done a good job in revising their manuscript, there are some remaining issues that I don’t feel have been dealt with satisfactory.

COMMENT 1: The first, and foremost, issue that I highlighted in the first review is that the authors make the statement that the paper “outline current best practice for drone operation in marine environments” (Lines 32-33). In my opinion this is unfair to state for all the experts in this field that has not had a chance to comment on this. These are recommended best practices by the authors of this paper, based on their experiences etc. (and the literature) and that needs to be made very clear to the reader. They are not speaking on behalf of an entire discipline. So please revise this and make it clear that these are recommendations from you, and not an “outline of current best practice”.

COMMENT 2: I don’t think the authors fully understood my previous comment (observe that line numbers refers to the previous version of the manuscript): “Lines 208-209: Water surface distortion can be a real issue when measuring body morphometrics of animals below the surface.” NEW COMMENT: I am talking about distortion caused by surface movement resulting from wind, the animal breaking the surface and creating spray etc. Are there ways to account for that?

COMMENT 3: “Lines 212-215: It would have been nice to see how errors could be incorporated into the measurements. For example if you measure 10 images from the same shark, perhaps the variation in body morphometric measurements could use used to obtain an error estimate of how accurate the measurement is.” NEW COMMENT: This comment does not refer to altimeter accuracy, but errors resulting from measuring a moving animal in the water. The posture of the animal, whether it is breaking the surface and causing distortion of the water etc. is a real issue that I don’t feel has been addressed in this review. Could you recommend ways of dealing with that? For these sorts of errors it does not matter how accurate your altimeter is, since that’s not the cause of the error.

COMMENT 4: “Lines 681: The study of Ramos et al. (2018) suffers from several limitations, including a lack of adequate control and also pseudo-replication. It should not be cited as there being evidence of dolphins reacting to drones.” PREVIOUS ANSWER BY THE AUTHORS: Respectfully, we disagree that this study should not be cited. Expecting controlled conditions in the wild to study animal behaviour is not realistic, and is actually more relevant than in controlled conditions (e.g. dolphinariums) where sound reverberation is characteristically different from natural conditions (see Duncan et al. 2016, covered for fishes in Popper and Hawkins 2019). In the absence of alternative studies, the Ramos et al. 2018 study is one of the few to explicitly attempt to measure reactions of dolphins to drones and it should be cited.

NEW COMMENT: I do not find this response from the authors satisfactory. Controlled experiments can indeed be done on wild cetaceans, as recently showed by Sprogis et al. (2020 Elife), although that’s not what I meant with “control” in this case. I also never suggested doing captive experiments of dolphins, so that was not the issues. What I meant with “control” is having data on undisturbed behaviour of dolphins collected without the drone or the boat being present and comparing this to data when the drone is present. That is not difficult to obtain and has been successfully collected in many behavioural impact studies on cetaceans and other taxa. Citing a study that suffers from a lack of control and also pseudo-replication (as well as having a potentially noisy boat close to the dolphins) should not be done, even if it’s the only study out there. So please either remove this citation or highlight the flaws of the study.

COMMENT 5: I still find “in prep.” articles being cited. Please remove these or replace with published work or personal communication. " I think if the authors satisfactory address the above issues then it is ready to be published. So I would suggest minor revision.

Author Response

Dear Editor,

Thank you for the opportunity to modify our manuscript for possible publication in Drones.  We thank the reviewers for their very helpful comments and the we believe the manuscript has been greatly improved. We have adopted the reviewer comments and made corresponding changes in the manuscript. Our responses to the reviewers, along with changes made, are in red font below.

_________________________________________________________________

The authors have done a good job in revising their manuscript, there are some remaining issues that I don’t feel have been dealt with satisfactory.

COMMENT 1: The first, and foremost, issue that I highlighted in the first review is that the authors make the statement that the paper “outline current best practice for drone operation in marine environments” (Lines 32-33). In my opinion this is unfair to state for all the experts in this field that has not had a chance to comment on this. These are recommended best practices by the authors of this paper, based on their experiences etc. (and the literature) and that needs to be made very clear to the reader. They are not speaking on behalf of an entire discipline. So please revise this and make it clear that these are recommendations from you, and not an “outline of current best practice”.

To avoid confusion, we have added 'our' to the sentence to make it clear that these best practices are based on the literature and our practical experience.

COMMENT 2: I don’t think the authors fully understood my previous comment (observe that line numbers refers to the previous version of the manuscript): “Lines 208-209: Water surface distortion can be a real issue when measuring body morphometrics of animals below the surface.” NEW COMMENT: I am talking about distortion caused by surface movement resulting from wind, the animal breaking the surface and creating spray etc. Are there ways to account for that?

COMMENT 3: “Lines 212-215: It would have been nice to see how errors could be incorporated into the measurements. For example if you measure 10 images from the same shark, perhaps the variation in body morphometric measurements could use used to obtain an error estimate of how accurate the measurement is.” NEW COMMENT: This comment does not refer to altimeter accuracy, but errors resulting from measuring a moving animal in the water. The posture of the animal, whether it is breaking the surface and causing distortion of the water etc. is a real issue that I don’t feel has been addressed in this review. Could you recommend ways of dealing with that? For these sorts of errors it does not matter how accurate your altimeter is, since that’s not the cause of the error.

Referring to comments 2 and 3, we feel this is already discussed explicitly lines 348 - 358. Using video when making photogrammetry can help select for frames where distortion (ie. position of the animal, wave clutter, spray, etc.) is minimal. See lines 353 - 355 in particular.

COMMENT 4: “Lines 681: The study of Ramos et al. (2018) suffers from several limitations, including a lack of adequate control and also pseudo-replication. It should not be cited as there being evidence of dolphins reacting to drones.” PREVIOUS ANSWER BY THE AUTHORS: Respectfully, we disagree that this study should not be cited. Expecting controlled conditions in the wild to study animal behaviour is not realistic, and is actually more relevant than in controlled conditions (e.g. dolphinariums) where sound reverberation is characteristically different from natural conditions (see Duncan et al. 2016, covered for fishes in Popper and Hawkins 2019). In the absence of alternative studies, the Ramos et al. 2018 study is one of the few to explicitly attempt to measure reactions of dolphins to drones and it should be cited.

NEW COMMENT: I do not find this response from the authors satisfactory. Controlled experiments can indeed be done on wild cetaceans, as recently showed by Sprogis et al. (2020 Elife), although that’s not what I meant with “control” in this case. I also never suggested doing captive experiments of dolphins, so that was not the issues. What I meant with “control” is having data on undisturbed behaviour of dolphins collected without the drone or the boat being present and comparing this to data when the drone is present. That is not difficult to obtain and has been successfully collected in many behavioural impact studies on cetaceans and other taxa. Citing a study that suffers from a lack of control and also pseudo-replication (as well as having a potentially noisy boat close to the dolphins) should not be done, even if it’s the only study out there. So please either remove this citation or highlight the flaws of the study.

We had already discussed this explicitly line 839, but have now specified that 'and thus results could be biased' line 839.

COMMENT 5: I still find “in prep.” articles being cited. Please remove these or replace with published work or personal communication. " I think if the authors satisfactory address the above issues then it is ready to be published. So I would suggest minor revision.

We have removed in prep citations throughout the document.

Reviewer 3 Report

The authors have done a very nice job in quickly addressing my comments. As I said in my first review, we need more of these types of papers to begin to set up standard/best practices and to help refine methods using UAS platforms to generate comparable science and data products. Nice work. 

Author Response

We thank the reviewer for their positive feedback.